# Identification of hidden N4-like viruses and their interactions with hosts

Kaiyang Zheng,[1,2] Yantao Liang,[1,2] David Paez-Espino,[3,4] Xiao Zou,[5] Chen Gao,[1,2] Hongbing Shao,[1,2] Yeong Yik Sung,[2,6] Wen Jye Mok,[2,6] Li Lian Wong,[2,6] Yu-Zhong Zhang,[1,7] Jiwei Tian,[8] Feng Chen,[9] Nianzhi Jiao,[10] Curtis A. Suttle,[11,12,13] Jianfeng He,[14] Andrew McMinn,[1,15] Min Wang[1,2,16]

**ABSTRACT**  The N4-like viruses, which were recently assigned to the novel viral family Schitoviridae in 2021, belong to a podoviral-like viral lineage and possess conserved genomic characteristics and a unique replication mechanism. Despite their significance, our understanding of N4-like viruses is primarily based on viral isolates. To address this knowledge gap, this study has established a comprehensive N4-like viral data sets comprising 342 high-quality N4-like viruses/proviruses (144 viral isolates, 158 uncultured viruses, and 40 integrated N4-like proviruses). These viruses were classified into 97 subfamilies (89 of which are newly identified), 148 genera (100 of which are newly identified), and 253 species (177 of which are newly identified). The study reveals that N4-like viruses inhibit the polar region, oligotrophic open oceans, and the human gut, where they infect various bacterial lineages, such as Alpha/Beta/Gamma/Epsilon-proteobacteria in the Proteobacteria phylum. Although N4-like viral endogenization appears to be prevalent in Proteobacteria, it has also been observed in Firmicutes. Additionally, the phylogenetic analysis has identified evolutionary divergence within the hallmark genes of N4-like viruses, indicating a complex origin of the different conserved parts of viral genomes. Moreover, 1,101 putative auxiliary metabolic genes (AMGs) were identified in the N4-like viral pan-proteome, which mainly participate in nucleotide and cofactor/vitamin metabolisms. Of these AMGs, 27 were found to be associated with virulence, suggesting their potential involvement in the spread of bacterial pathogenicity.

**IMPORTANCE**  The findings of this study are significant, as N4-like viruses represent a unique viral lineage with a distinct replication mechanism and a conserved core genome. This work has resulted in a comprehensive global map of the entire N4-like viral lineage, including information on their distribution in different biomes, evolutionary divergence, genomic diversity, and the potential for viral-mediated host metabolic reprogramming. As such, this work significantly contributes to our understanding of the ecological function and viral-host interactions of bacteriophages.

**KEYWORDS**  N4-like viruses, genome, diversity, evolutionary divergence, viral-mediated host metabolic reprogramming

The *Escherichia* virus N4 was first isolated from sewage in 1966 by Gian Carlo Schito and has a podoviral morphology (1). This virus is characterized by the giant virion-encapsulated RNA polymerase (N4-vRNAP, N4-gp50), which is over 10 kbp in length. In the post-infection period, lysis inhibition is often observed in *Escherichia* virus N4, leading to an extended latent period and a huge burst size (about 3,000 PFU/cell) (2, 3). N4-like viruses can lytically infect many pathogenic bacteria and have shown efficient therapy performance in previous animal experiments, making them potential candidates for phage therapy of multidrug-resistant pathogenic bacteria (4–9). Many

Address correspondence to Yantao Liang, liangyantao@ouc.edu.cn, Jianfeng He, hejianfeng@pric.org.cn, Andrew McMinn, andrew.mcminn@utas.edu.au, or Min Wang, mingwang@ouc.edu.cn.

Kaiyang Zheng, Yantao Liang, and David Paez-Espino contributed equally to this article. Author order was determined in order of increasing seniority.

The authors declare no conflict of interest.

See the funding table on p. 18.

N4-like viruses can infect pathogenic or opportunistically pathogenic bacteria, such as *Escherichia*, *Achromobacter*, *Shigella*, *Vibrio*, *Klebsiella*, and *Pseudomonas* (4, 7, 9–12). In 2021, the International Committee on Taxonomy of Viruses (ICTV) assigned this special viral lineage to the novel viral family Schitoviridae, named in memory of Gian Carlo Schito (13).

As of January 2023 (the time of this writing), a total of 144 complete and non-redundant N4-like viral genomes have been published in GenBank. N4-like viruses are known to colonize widely in eutrophic environments and infect diverse hosts. Genomic and taxonomic studies of N4-like viral isolates have demonstrated the relatively close pangenome features of N4-like viruses (14, 15). Only one study reported on the diversity and geographic distribution pattern of N4-like viruses, identifying their preference for cold water (16). However, our understanding of N4-like viruses is still limited and mainly based on viral isolates. A lack of a global contextualized map of uncultured N4-like viruses means that the diversity of this viral lineage is still underestimated.

Environmental metagenomics has greatly expanded our knowledge of viral diversity, and many metagenome-based databases have been established, such as the Integrated Microbial Genome/Metagenome and Gut Virome Database (GVD) (17, 18). Paez-Espino and colleagues determined over 1.2 million viral genomes from over 3,000 environmental metagenomes (19). These uncultured viral genomes (UViGs) were included in the Integrated Microbial Genome/Viral Resource (IMG/VR), the largest uncultured DNA viral database at present (20). Based on UViGs, in-depth study of specific viral lineages has provided many unique viral genomic and phylogenetic characteristics. For instance, Weinheimer and Aylward discovered 97 bacteriophages encoding multi-subunit (ββ′) RNA polymerase, which represents an ignored and widely branching viral lineage (21). Kieft and his colleagues discovered 191 bacteriophages that potentially participate in bacterial inorganic sulfur metabolism, providing evidence of viral-mediated sulfur biogeochemical cycles (22). We speculate that this powerful method could improve our understanding of the importance of uncultured N4-like viruses.

In this study, we confirmed a total of 795 N4-like viral genomes from GenBank (2022/08) and the IMG/VR (v.3) using data-mining and filtering steps. This data set includes 342 N4-like high-quality viral genomes [144 viral isolates, 158 uncultured viruses (high-quality uncultivated viral genomes, HQ-UViGs), and 40 integrated N4-like proviruses] (see Table S1 at https://bitbucket.org/minwanglab/n4-like_viruses_supplementary_materials). These HQ-UViGs were widely distributed in many natural ecosystems and host-associated environments, and the N4-like viral lineage now includes 97 subfamilies, 148 genera, and 253 species encoding different types of auxiliary metabolic genes (AMGs). It was also found that the different hallmark genes of N4-like viruses might be involved in different evolutionary processes. The discovery of integrated N4-like proviruses provides evidence of lysogenic life strategies in some N4-like viruses. Overall, this study offers new insights into the habitat, phylogeny, genomics, and functional diversity of N4-like viruses.

## RESULTS AND DISCUSSION

### Genomic features of N4-like UViGs from diverse environmental metagenomes

Following a systematic and comparative genomic analysis of N4-like viral genomic sequences in GenBank, this study identified seven hallmark genes, including viral-encoded DNA polymerase (N4-vDNAP, N4-gp39, SchitoPF00003), major capsid protein (N4-MCP, N4-gp56, SchitoPF00005), portal protein (N4-Portal, N4-gp59, SchitoPF00006), tail protein (N4-Tail, N4-gp67, SchitoPF00007), terminase large subunit (N4-TerL, N4-gp68, SchitoPF00008), terminase small subunit (N4-TerS, N4-gp69, SchitoPF00009), and virion-encapsulated RNA polymerase (N4-vRNAP, N4-gp50, SchitoPF00019). These hallmark genes were used to search the NR database (2022/08) and IMG/VR (v.3) using a series of data-mining and filtering steps, resulting in the identification of 158 N4-like HQ-UViGs out of 611 N4-like UViGs from 141 metagenomes with diverse

environmental features. The 158 N4-like HQ-UViGs were found in various environments, with most of them detected in aquatic-associated metagenomes (marine/sediment, freshwater/deep subsurface, and non-marine saline and alkaline, $n = 76$), followed by host-associated metagenomes (human/mammals, annelida/arthropoda, phyllosphere, algae, and sphagnum, $n = 66$), terrestrial-associated metagenomes (terrestrial/soil, $n = 9$), and engineered-associated metagenomes (wastewater and solid waste/bio-transformation/built environment, $n = 7$) (Fig. 1 and 2A, see Table S1 at https://bitbucket.org/minwanglab/n4-like_viruses_supplementary_materials). Furthermore, 40 integrated, high-quality N4-like proviral regions were identified in 40 bacterial genomic contigs, mainly from *Moraxella* ($n = 28$). Additionally, the study found that N4-like viruses can completely integrate into the genomes of *Enterococcus faecium* (strains QAUEFNN4 and ME3). To the best of our knowledge, if not considered an artifact of chimeric assemblies, this provides the first evidence that N4-like viruses can infect Firmicutes lineages in a lysogenic way.

The genomic lengths of N4-like HQ-UViGs in the four main ecosystems were similar, especially in the three non-host-associated ecosystems (Fig. 2B). The genomic lengths of HQ-UViG ranged from 42 to 131 kbp, with a median and average assembled length of 73 kbp, which is similar to the N4-like viral isolates (23). The medians of assembled lengths of N4-like HQ-UViGs from these main ecosystems are similar (48 kbp for aquatic, 54 kbp for terrestrial-associated, and 51 kbp for engineered-associated), except for the host-associated N4-like UViGs (69 kbp), suggesting that the N4-like UViGs from

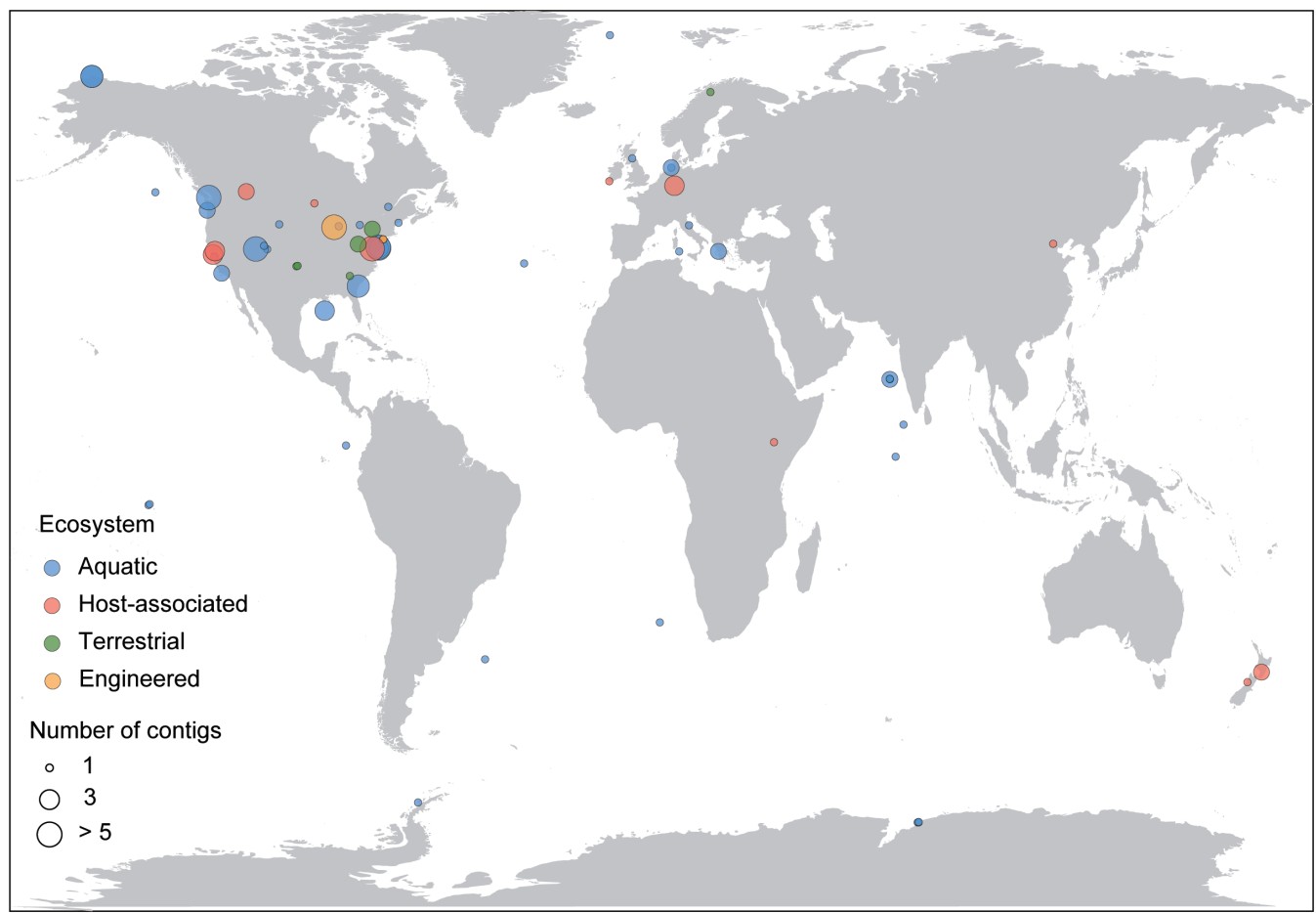

**FIG 1** Geographical and biome distribution of N4-like viral sequences detected in metagenomes included in IMG/VR (v.3). Each sample is represented by a circle proportional to the number of high-quality N4-like viral genome detections and colored according to their ecosystem type. These high-quality N4-like viral genomes were from 138 metagenomes, including 56 from host-associated, 50 from marine, 12 from freshwater, 9 from terrestrial, 7 from engineered, 3 from non-marine saline and alkaline, and 1 from sediment.

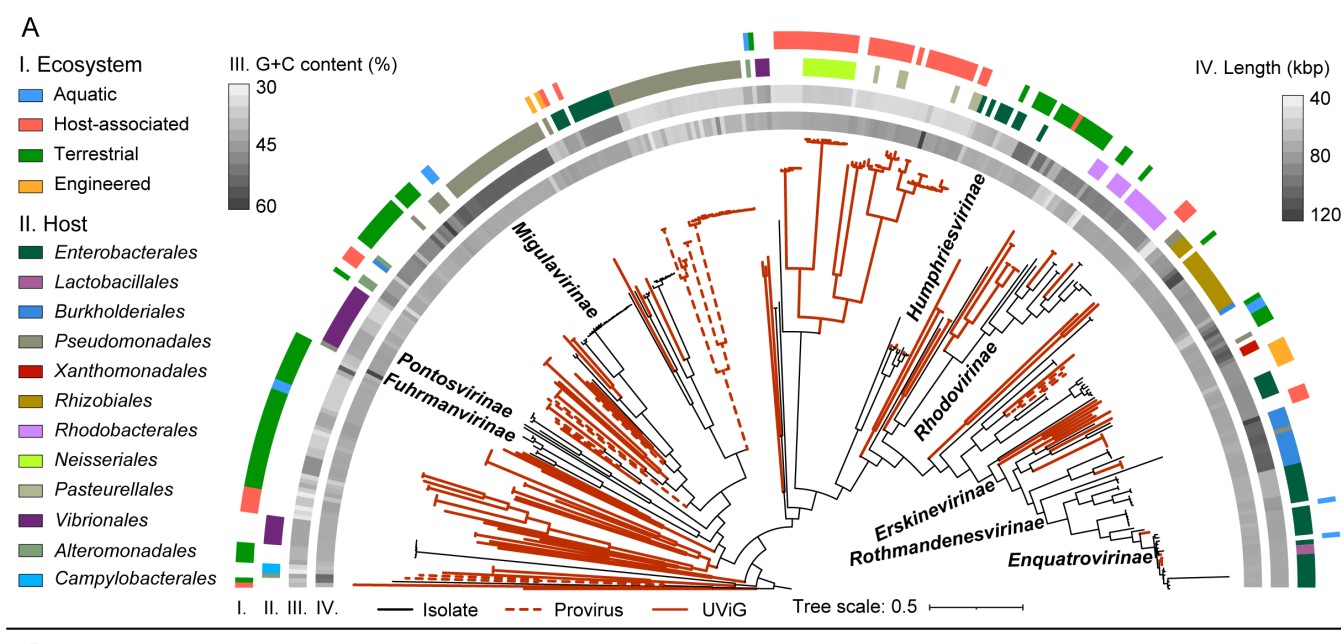

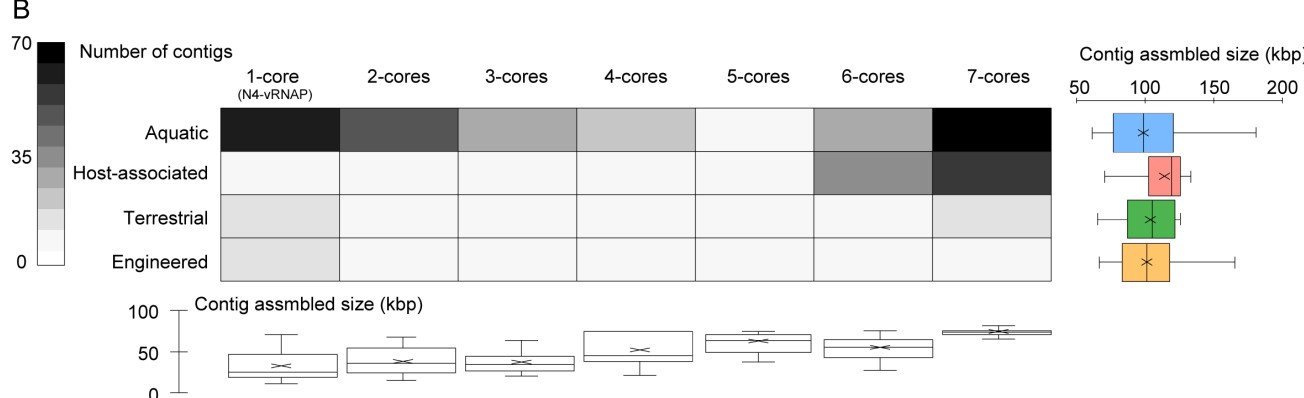

**FIG 2** Metagenomic expansion of the N4-like viral diversity. (A) Maximum-likelihood phylogenetic tree of the N4-like viruses inferred from a concatenated protein alignment of seven hallmark genes. Branches in red solid/dashed lines represent HQ-UViGs and proviruses, respectively. Tree annotations from outside to inside: I, metagenome environmental origin; II, experimental host or predicted lineages of N4-like viral isolates; III, assembly size of N4-like viral genomes; IV, percentage of G + C content of N4-like viral genomes. (B) The heatmap shows the number of contigs along with the number of hallmark gene accumulations in four main ecosystem types. The accumulation of one hallmark gene (1-core) contains genomes encoding only N4-like virion-encapsulated RNA polymerase (N4-vRNAP, N4-gp50, SchitoPF00019). The categories of 1-core to 7-core contained 287, 56, 26, 22, 9, 51, and 342 contigs, respectively. The box charts on the right and below the heatmap indicate the genome assembly size in four ecosystem types and in different numbers of hallmark gene accumulations, respectively.

host-associated environments tend to have a higher completeness of genomic assembly. The assembly length generally increases with the accumulation of hallmark genes (Fig. 2B) and strongly converges when the number of hallmark genes reaches seven. Most of the N4-like UViGs encoding the seven hallmark genes were HQ-UViGs, except for two artificially concatenated sequences (artifacts from the assembly process introduced by the greedy algorithm), which contained multiple copies of the hallmark genes as artifacts from the assembly process (20). Therefore, only N4-like UViGs encoding all seven single-copy hallmark genes were regarded as HQ-UViGs.

## Pangenome, taxonomic, and host spectrum expansion of N4-like viruses

Although previous works have made some efforts to recognize N4-like viruses containing a conserved core genome (14, 15, 23), an unbiased view of the N4-like viral pangenome is still lacking due to the limited sample size. In this study, we performed a

systematic pangenome analysis for N4-like viruses based on 342 high-quality N4-like viral genomes (see Table S1 at https://bitbucket.org/minwanglab/n4-like_viruses_sup-plementary_materials). A total of 35,044 viral proteins involved in 1,046 protein families (SchitoPFs) were identified, including 17,968 and 1,807 viral proteins from identified N4-like HQ-UViGs and proviruses, respectively (see Extending Data 1 at https://bit-bucket.org/minwanglab/n4-like_viruses_supplementary_materials). This work doubled the diversity of N4-like viral proteins compared with isolates. The accumulation curve of viral SchitoPFs was proportional to the accumulation of viral genomes (Fig. 3B), indicating that the genomic diversity of N4-like viruses had not been fully identified.

Genome-content-based networks and intergenomic similarity clustering were used to expand the taxonomic scope of N4-like viruses. From the genome-content-based network, 29 viral clusters (VCs) and 24 viral singletons (SGs) were generated (Fig.

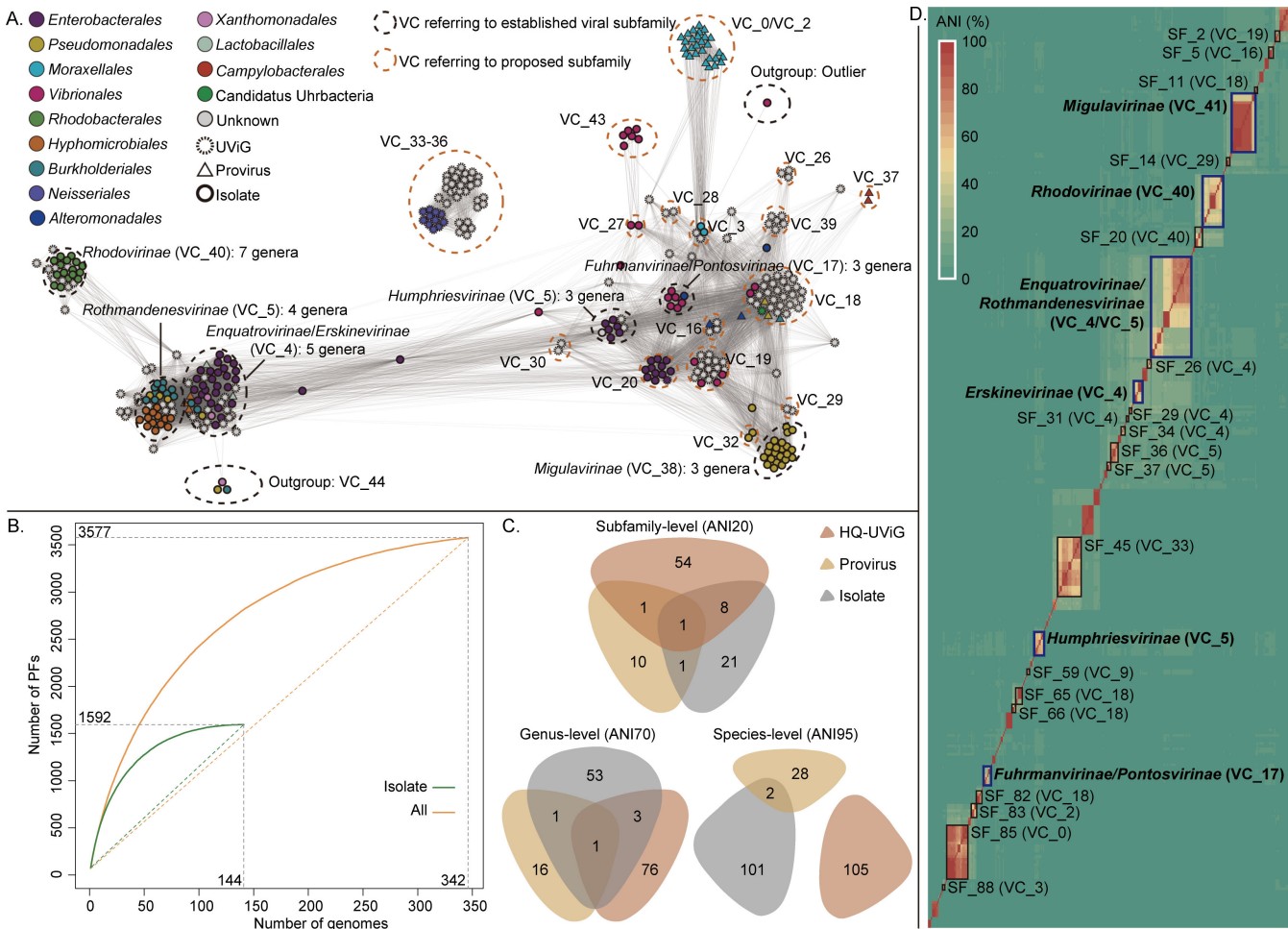

**FIG 3** The pangenome feature and classification of N4-like viruses. (A) The genome-content-based network of N4-like viruses. Nodes with different colors represent a high-quality N4-like viral genome and corresponding host lineage. Related VCs and established subfamilies are indicated beside clusters. The VCs of the proposed subfamilies in this study are enclosed by dashed lines. (B) The PF accumulation curve of the expanded N4-like viral pan-proteome. This curve is linked by the medians of each subsampled databox (300-time random sampling for each data point; the box and error bar are not shown here). The curves in orange and green represent PF accumulation of 144 N4-like viral isolates and all 342 high-quality N4-like viral genomes, respectively. The dashed lines linking the ends of curves represent the slopes of two curves, reflecting the average accumulation rates of PFs accompanied by the viral pan-proteome expansion. (C) The viral species (ANI ≥95%), genus (ANI ≥70%), and subfamilies (ANI ≥20%) overlap between N4-like viral isolates, HQ-UViGs, and proviruses. The related VCs and corresponding established/proposed subfamilies were determined by the VIRIDIC-based ANI heatmap (D). (D) The VIRIDIC-based ANI heatmap shows the established subfamilies (enclosed by boxes in bold blue lines) and the proposed subfamilies (enclosed by boxes in bold blue lines; the proposed subfamilies containing single members are not shown). Members within each established/proposed subfamily (except Enquatrovirinae/Rothmandenesvirinae and Fuhrmanvirinae/Pontosvirinae) have at least 20% ANI. This is a minimally conservative estimate result (see Materials and Methods).

3A and D; see Table S1 at https://bitbucket.org/minwanglab/n4-like_viruses_supplementary_materials). The 29 VCs belong to 97 subfamilies, including 89 manually identified proposed subfamilies based on the clustering of genome-wide average nucleotide identity. The largest subfamily is Enquatrovirinae (part of VC4, including 26 viruses), which is mainly composed of isolated N4-like viruses and mainly serves Enterobacteriaceae as their hosts. The Enquatrovirinae include two integrated proviruses identified from two genomic contigs of *Enterococcus* (strains QAUEFNN4 and ME3), both of which are *Gamaleyavirus PGN829.1* and also infect *Escherichia*, implying the potential cross-phyla horizontal transfer within the microbiome mediated by N4-like viruses. The largest proposed subfamily is *SF45* (VC33, including 22 viruses), which is all composed of HQ-UViGs. The CRISPR-spacer matching shows that *SF45* serves *Haemophilus* as its host (24). The *SF45* were clustered with the other three proposed subfamilies (*SF42*, *SF43*, and *SF47*), forming another super cluster outside the main network. From the host prediction (CRISPR-spacer matching and consensus vOTU host) (19, 25), *SF43* serves *Neisseria* as its host. In the concatenated tree of hallmark genes, this superclade also formed a deeply monophyletic branch (Fig. 2A). This result suggests a hidden N4-like viral lineage, which still lacks any culturing isolates. In addition, the N4-like proviruses from genomes of *Moraxella* were classified into three proposed subfamilies (*SF83*, *SF84*, and *SF85*) and clustered as two VCs associating with VC_3 (including three proposed subfamilies and two defined genera, *Exceevirus* and *Presleyvirus*) in the network (Fig. 3A and D). The *SF83*, *SF84*, and *SF85* are only composed of N4-like proviruses and form a deeply monophyletic branch in the concatenated tree of hallmark genes (Fig. 2A). This implies that these N4-like viral endogenizations may drive divergences in both genome content and phylogeny compared with other free-living viruses.

Overall, this result remarkably expanded the pangenome content and genomic catalog of N4-like viruses, suggesting that the isolated N4-like viruses are only the tip of the iceberg. The N4-like viruses infect at least 14 bacterial orders, while isolated viruses are only from 8 bacterial orders. These uncultured N4-like viruses might widely serve Neisseriales, Pasteurellales, Campylobacterales, Rhodocyclales, Oceanospirillales, and Lactobacillales as their natural hosts (Fig. 3A and see Table S1 at https://bitbucket.org/minwanglab/n4-like_viruses_supplementary_materials). Only limited diversity overlaps between N4-like viral isolates (Fig. 3C), HQ-UViGs, and integrated proviruses, indicating that the diversity of N4-like viruses is more complex than we acknowledged before.

## Evolutionary of N4-like viruses

The DNA-packaging-related protein is highly conserved in dsDNA viruses and has been used to study the evolution of N4-like viruses and other dsDNA viruses (23, 26–29). N4-like viruses constitute a viral group that forms a deeply branching phylogenetic clade in the virosphere, indicating their conserved evolutionary status compared to other viral families and genera within Caudoviricetes (Fig. 4A) (30, 31). The phylogeny of N4-TerL and other homologous proteins in viruses reveals that all investigated N4-like viruses cluster together, forming a monophyletic clade, which agrees with the study conducted by Wittmann et al. (23). The Autographiviridae (T7-like viruses) display closer phylogenetic relationships with N4-like viruses than other outgroups, suggesting that both viral families might have evolved from a common ancestor. In the phylogenic tree of N4-TerL, each subfamily has relatively conserved branches and is specific to its host lineage, indicating that the evolution of N4-like viruses may correspond to their sensitive hosts (Fig. 4B). However, this result does not agree with the multigene concatenated tree, such as N4-like viruses infecting Enterobacterales and Vibrionales, indicating that other hallmark genes of N4-like viruses may have undergone horizontal transfer events between different N4-like viral subfamilies (Fig. 2A), resulting in the phylogenetic divergence between two trees.

Other hallmark genes of N4-like viruses, such as the phylogeny of N4-vRNAP, N4-vDNAP, and N4-TerL, have also been used to infer the classification of N4-like

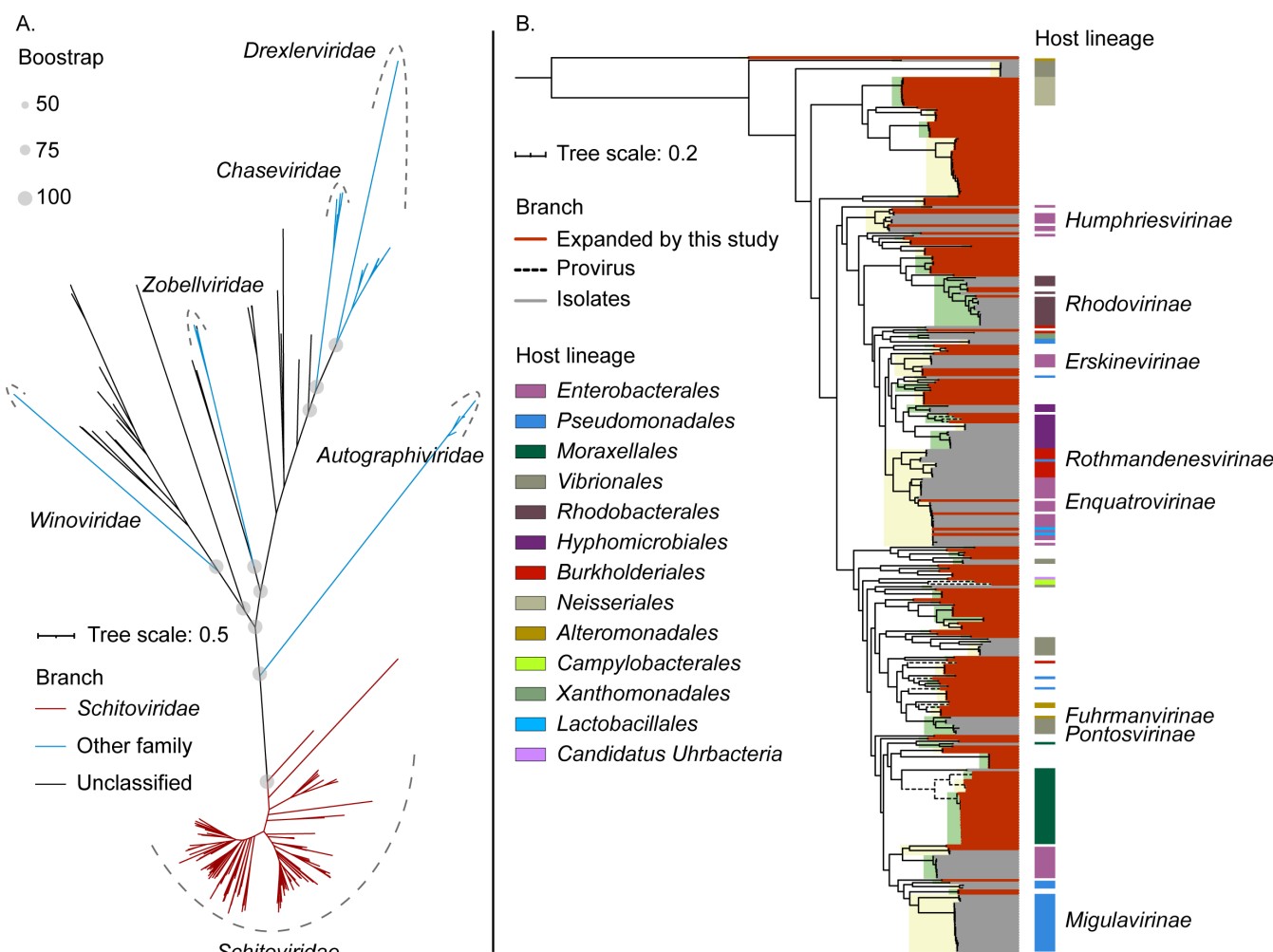

**FIG 4** The phylogenic trees of N4-like viruses based on N4-like terminase large subunits (N4-TerL, N4-gp68, SchitoPF00008). (A) The maximum-likelihood tree of N4-like viruses and other related caudoviruses. Branches in red represent all 342 high-quality (HQ) N4-like viral isolates/UViGs/proviruses; branches in blue represent other viruses from established families; branches in black represent other viruses within floated genera (currently pending as taxonomy proposals). (B) The expanded subtree of the Schitoviridae clade from the global tree (A). Branches labeled in red lines represent expanded N4-like viruses from this study, including HQ-UViGs (solid lines in black) and proviruses (dashed lines in black). The tree annotations represent the corresponding experimental host lineages or predicted host lineages of N4-like viruses.

viruses (14, 16, 32–34). Previous studies have shown that the N4-like viral hallmark genes might have co-evolved. However, the quantitative description of phylogenetic heterogeneity/homogeneity between N4-like hallmark genes is still lacking (26, 35). As a result of the co-phylogenic analysis based on seven hallmark genes, the phylogenies of single hallmark genes were not always consistent with the phylogenies of the multigene (Fig. 5A). The phylogeny of the multigene was mostly correlated to the phylogenies of N4-Portal and N4-vRNAP (strict Robinson-Foulds values (sRFVs) = 0.477 and 0.354, respectively). The N4-Portal is also highly correlated to the N4-vRNAP (sRFV = 0.395). This is reflected in the co-phylogenetic tree of N4-Portal-vRNAP, where the topology of the two trees is highly similar (Fig. 5B). Three main deeply branching clades were generated for both trees and were composed of similar N4-like viral subfamilies (except SF20). However, the phylogenetic tree topologies of N4-vDNAP and N4-TerS show divergence, with only two clades comprising the same subfamilies being identical (Fig. 5C). Given the phylogenetic heterogeneity observed in different N4-like hallmark genes, the N4-like hallmark genes might have diverse origins and might have been vertically transferred from different ancestors to form the current N4-like viral core genome. Similar to

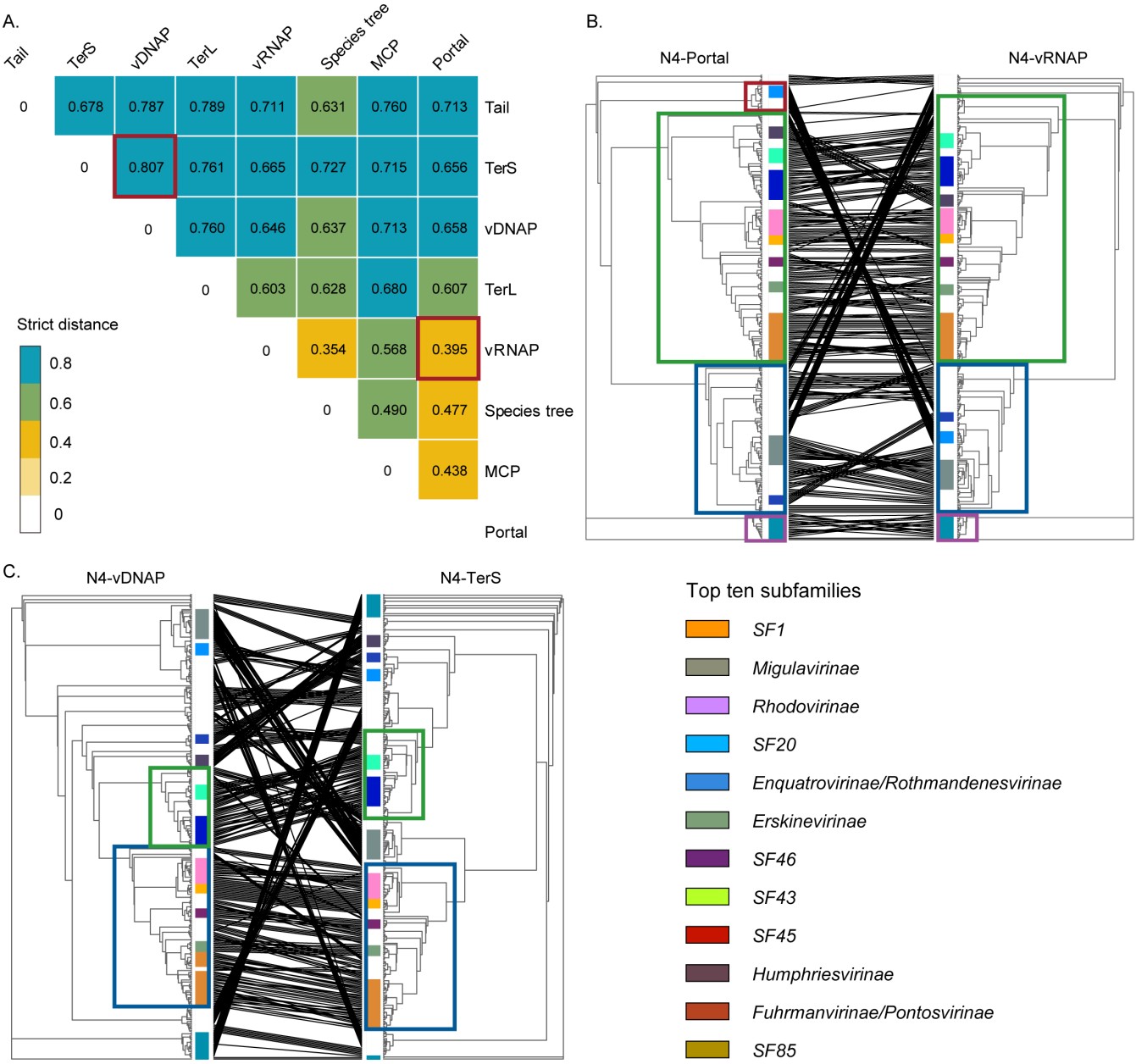

**FIG 5** The phylogenic homogeneity and heterogeneity of N4-like viral hallmark genes. (A) The sRFV heatmap describes the topology similarities of the maximum-likelihood tree between N4-like viral hallmark genes and between seven hallmark gene concatenated core genome. The sRFVs are closer to 0, the more similar the tree topology. The max and min of sRFVs are highlighted by boxes in red. The corresponding co-phylogenic trees are shown in panels (B and C), respectively. Panels (B and C) are co-phylogenic trees of N4-Portal/N4-vRNAP (the hallmark gene pair with the most similarity of tree topology) and N4-vDNAP/N4-TerS (the hallmark gene pair with the least similarity of tree topology). The nearly identical clades between two trees are highlighted by boxes in the same color, consisting of similar subfamily-level units. To make the figure clearer, only the top 10 subfamilies with the highest member counts are displayed. Abbreviations: vDNAP, viral DNA polymerase; vRNAP, virion-encapsulated RNA polymerase; TerL, terminase large subunit; TerS, terminase small subunit; MCP, major capsid protein.

the Zobellviridae with a hybrid podo-autographic genotype (36–38), this hybrid core genome may give selective advantage to N4-like viruses in the evolutionary process.

In addition to the co-phylogeny of N4-like viruses, we assess the residue-level covariance between these hallmark genes on the basis of the study of Wangchuk et al. (26). The Direct Coupling Analysis with mean-field approximation (mfDCA) coevolutionary algorithm was used to describe the quantitative covariance between N4-like viral

hallmark genes (26). Six hallmark gene pairs with potential covariance were screened out compared with the negative control (TerL and integrase of *Lambdavirus*) and positive control (TerL and portal protein of *Enquatrovirus* N4), including Portal-Tail, Portal-TerS, TerL-Tail, TerL-TerS, vDNAP-Tail, and vDNAP-TerS (Fig. 6; see Table S2 at https://bit-bucket.org/minwanglab/n4-like_viruses_supplementary_materials). The mfDCA values from these pairs differ significantly from the negative control. The covariance is frequently observed within viral late genes, while N4-vDNAP is the only viral early gene

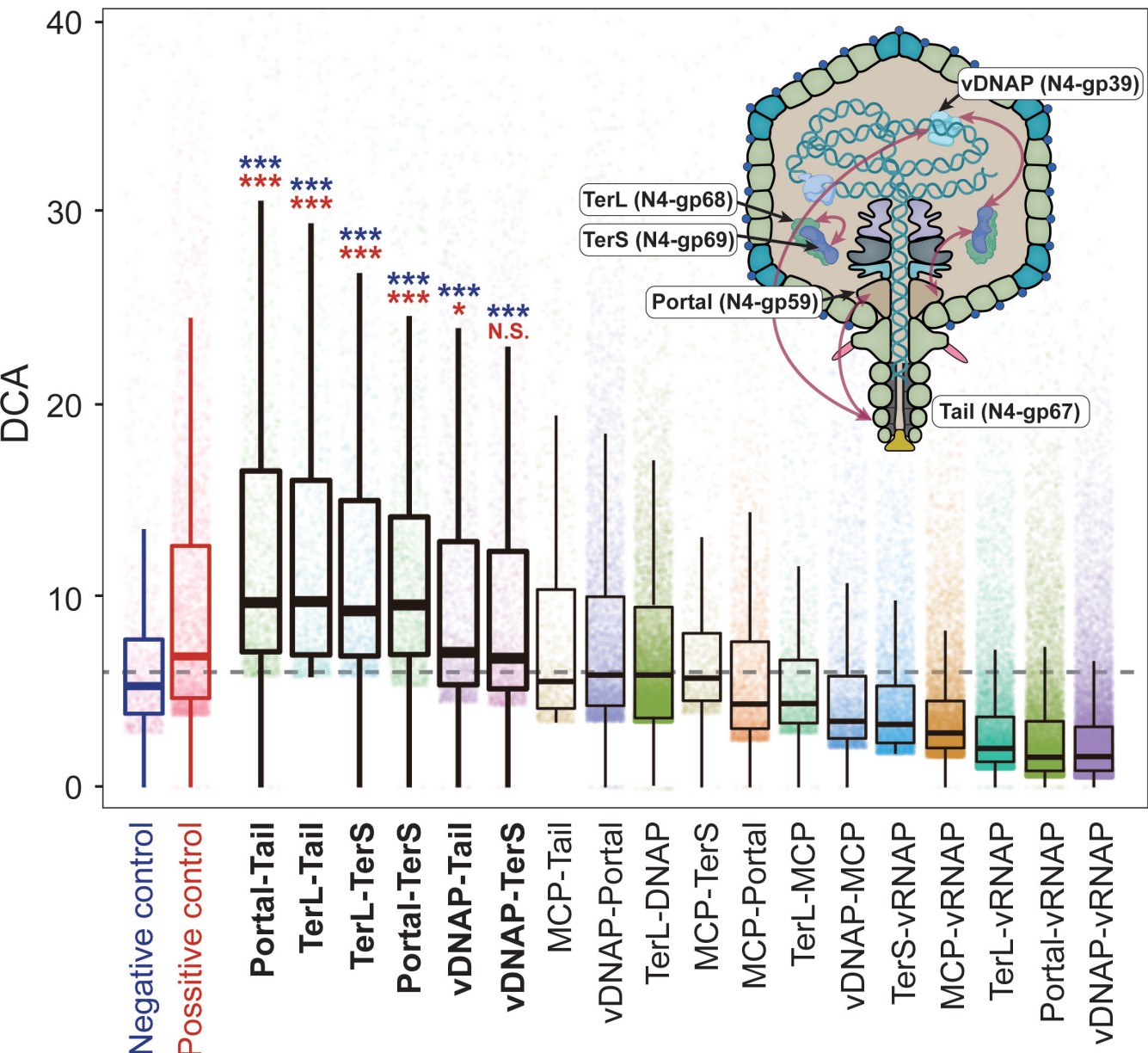

**FIG 6** Amino acid level covariance of N4-like viral hallmark genes. The TerL-Int and TerL-Portal encoded by *Lambdavirus Lambda* were used as negative and positive controls, respectively. The databoxes represent the top 1% mfDCA, which were used to describe the covariance characterization. Protein pairs were considered to have high covariance if their medians of mfDCA values were higher than the medians of positive and negative controls. The differences in significance between protein pairs and controls were checked though a *t*-test. ***, *, and N.S. represent statistically significant differences with the negative/positive controls, $P < 0.001$, 0.05, and no significance, respectively. * in blue and red represents the comparison between negative/positive controls, respectively. The conceptual diagrams of covariance between N4-like viral hallmark proteins in the N4-like virion. The two viral hallmark proteins connected by arrows represent a pair of covariance components. Abbreviations: vDNAP, viral DNA polymerase; vRNAP, virion-encapsulated RNA polymerase; TerL, terminase large subunit; TerS, terminase small subunit; MCP, major capsid protein.

in the covariance network. This implies that the replication process of N4-like viruses might be conserved with other late processes. Considering the phylogenetic heterogeneity that occurred between N4-vDNAP and the other two late genes (Tail and TerS, reflecting the relatively large sRFVs compared with other pairs) (Fig. 5A), the N4-vDNAP, N4-Tail, and N4-TerS may have different origins but be conserved and coordinated in biological functions. The covariance partner of N4-vRNAP was not observed, implying that the N4-vRNAP-controlled early gene transcription process might be a relatively independent biological process (39, 40).

## Host metabolic reprogramming potential and putative lysogenic features of N4-like viruses

Metabolic reprogramming is a well-known phenomenon in which bacteriophages acquire AMGs from their hosts and express these genes during infection (41). Based on the genomic diversity expansion, this study provides comprehensive information on the host metabolism reprogramming potential and possible living strategies of N4-like viruses (Fig. 7; see Table S3 at https://bitbucket.org/minwanglab/n4-like_viruses_supplementary_materials). A total of 1,101 potential AMGs were

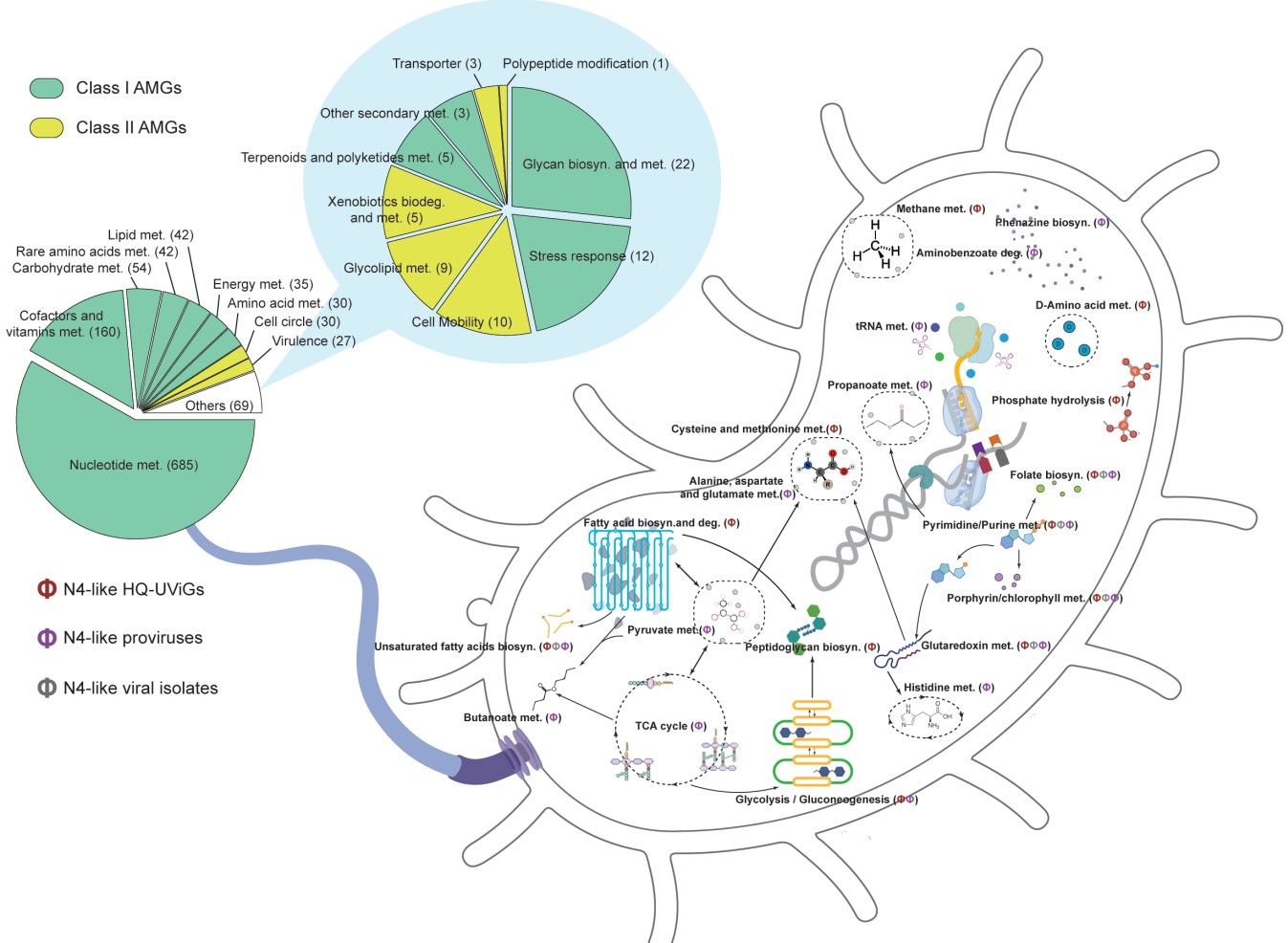

**FIG 7** The conceptual diagrams of host metabolism reprogramming for N4-like viruses. The N4-like viral proposed AMGs were detected based on the KO and Pfam-A databases. These AMGs were assigned into two classes: 18 main metabolic categories and 216 functional types. The detailed statistical information is shown by the pie charts. The diagrammatic sketches of major metabolic pathways involved N4-like viruses [N4-like HQ-UViGs, N4-like viral isolates, and N4-like proviruses]. To make it clear, only the top five pathways involved in three categories of viruses as well as the top five unique pathways for each category were shown in the diagram. Abbreviations: met, metabolism; biosyn, biosynthesis; biodeg, biodegradation.

identified from the N4-like pan-proteome against the KEGG Orthologs (KO) and Pfam-A databases, including 1,009 Class I AMGs and 92 Class II AMGs (42). These AMGs were classified into 18 metabolic categories and 90 different pathways (see Table S3 at https:// bitbucket.org/minwanglab/n4-like_viruses_supplementary_materials). Of the AMGs, 433 were encoded by N4-like viral isolates, 543 were encoded by HQ-UViGs, and 125 were encoded by N4-like proviruses. Therefore, this study expanded the number of N4-like viral AMGs by over 2.5 times. The majority of AMGs were associated with nucleotide metabolism ($n = 685$) and the metabolism of cofactors/vitamins ($n = 160$). The remaining AMGs were associated with carbohydrate metabolism ($n = 54$), metabolism of rare amino acids ($n = 42$), and lipid metabolism ($n = 42$) (Fig. 7). This result suggests that N4-like viruses can hijack and alter the pathways and efficiency of host energy and protein production (43).

Some AMGs of N4-like viruses are stable, while others are highly diverse, depending on their habitats. Nucleotide metabolism-related AMGs were stably encoded by N4-like viruses, regardless of the subfamilies or the habitats from which they originated. Each N4-like virus harbored an average of 2–3 nucleotide metabolism-related AMGs, which were carried by 27 subfamilies (Enquatrovirinae, Pontosvirinae, Migulavirinae, and 24 proposed subfamilies) (Fig. 7; Table 1; see Table S3 at https://bitbucket.org/minwan-glab/n4-like_viruses_supplementary_materials). This suggests that nucleotide metabolism and host reprogramming might be key biological processes for the infection of N4-like viruses. In contrast, cofactor/vitamin metabolism-related AMGs tended to be carried by N4-like viruses from host-associated habitats, while cell circle-related AMGs are rare in such habitats. Cell circle-related AMGs tended to be carried by N4-like viruses from natural environments, including marine, non-marine saline and alkaline, freshwater, terrestrial, and sediment (Table 1). This suggests that the host metabolism reprogramming potential of N4-like viruses varies by their habitat.

Importantly, virulence-associated AMGs were commonly harbored by N4-like proviruses ($n = 27$) and encoded by N4-like viruses from host-associated and non-marine saline and alkaline habitats (Fig. 7; Table 1). These AMGs include YadA-like membrane anchor domain (PF03895), virulence-associated protein E (PF05272), YopX protein (PF09643), and pre-toxin domain with VENN motif (PF04829). The AMGs of the YadA-like

**TABLE 1** The average number of auxiliary metabolic gene (AMG) encoding for different types of N4-like viruses from different habitats[a]

| Functional catagory | Engineered | H.A. | Marine | N.M.S.A. | Freshwater | Terrestrial | Sediment | UViG | Isolate | Provirus |
|---|---|---|---|---|---|---|---|---|---|---|
| Nucleotide metabolism | 2.86 | 2.94 | 2.84 | 2.5 | 2.23 | 1.56 | 0.33 | 2.81 | 2.61 | 2.58 |
| Metabolism of cofactors and vitamins | 0.71 | 2 | 0.42 | 0 | 0.54 | 0 | 0 | 1.08 | 0.35 | 0.6 |
| Cell circle | 0 | 0.02 | 0.18 | 0.25 | 0.23 | 0.22 | 0.33 | 0.12 | 0.06 | 0.05 |
| Carbohydrate metabolism | 0 | 0.08 | 0.06 | 0 | 0.92 | 0 | 0 | 0.13 | 0.1 | 1 |
| Glycan biosynthesis and metabolism | 0 | 0 | 0.05 | 0 | 0.15 | 0 | 0.67 | 0.04 | 0.12 | 0 |
| Metabolism of other amino acids | 0 | 0.59 | 0.06 | 0 | 0.08 | 0 | 0 | 0.28 | 0.03 | 0.18 |
| Virulence | 0 | 0.24 | 0.02 | 0.25 | 0.08 | 0 | 0 | 0.12 | 0.01 | 0.15 |
| Energy metabolism | 0 | 0.06 | 0.27 | 0.25 | 0 | 0 | 0 | 0.14 | 0.12 | 0.18 |
| Lipid metabolism | 0 | 0.02 | 0.23 | 0 | 0.08 | 0 | 0 | 0.1 | 0.19 | 0.2 |
| Metabolism of terpenoids and polyketides | 0 | 0 | 0 | 0.25 | 0 | 0 | 0 | 0.01 | 0 | 0.13 |
| Amino acid metabolism | 0 | 0.08 | 0.1 | 0 | 0.08 | 0 | 0 | 0.08 | 0.11 | 0.4 |
| Xenobiotics biodegradation and metabolism | 0 | 0 | 0 | 0 | 0.23 | 0 | 0 | 0.02 | 0 | 0.38 |
| Glycolipid metabolism | 0 | 0.02 | 0.08 | 0 | 0.08 | 0 | 0 | 0.04 | 0.01 | 0 |
| Transporter | 0 | 0 | 0 | 0 | 0 | 0.11 | 0 | 0.01 | 0 | 0.05 |
| Cell mobility | 0 | 0.05 | 0.02 | 0 | 0 | 0 | 0 | 0.03 | 0.03 | 0.03 |
| Polypeptide modification | 0 | 0 | 0.02 | 0 | 0 | 0 | 0 | 0.01 | 0 | 0 |
| Stress response | 0 | 0 | 0.02 | 0 | 0 | 0 | 0 | 0.01 | 0.06 | 0.08 |
| Biosynthesis of other secondary metabolites | 0 | 0 | 0 | 0 | 0 | 0 | 0 | 0 | 0 | 0.18 |
| DGR | 0 | 0 | 0.06 | 0.25 | 0 | 0 | 0 | 0.03 | 0 | 0 |
| Phage integrase | 0 | 0.06 | 0.15 | 0 | 0.08 | 0 | 0 | 0.09 | 0 | 0.73 |

[a]These values were calculated based on the total number of AMGs detection in certain categories normalized by the total number of N4-like viral genome detections in certain habitats. (H.A., Host-Associated; N.M.S.A., Non-Marine Saline and Alkaline; UViG, uncultured viral genome).

membrane anchor domain were homologous with *Yersinia* adhesin and were the most abundant virulence-associated AMGs (44). It was encoded by 19 genomes, including 2 N4-like viral isolates (*Waedenswilvirus vB_EamP-S6* and *Triduovirus vB_SalP_TR2*) infecting pathogenic bacteria in plants and humans, respectively (45, 46). The YadA-like membrane anchor domain is the core domain in the trimeric autotransporter adhesins of gram-negative bacteria membranes that are involved with the adhesion process with multicellular hosts of pathogens (47). Hence, it is speculated that the corresponding N4-like viruses harboring the YadA-like membrane anchor domain (PF03895) may increase the pathogenicity of related bacteria. The N4-like HQ-UViGs encoding YadA-like membrane anchor domains all belong to Humphriesvirinae, *SF39*, *SF45*, and *SF47*. Three proposed subfamilies are all from host-associated habitats (see Table S3 at https:// bitbucket.org/minwanglab/n4-like_viruses_supplementary_materials). *SF45* is closely associated with *Haemophilus parainfluenzae* through viral host prediction (see Table S1 at https://bitbucket.org/minwanglab/n4-like_viruses_supplementary_materials). As *Haemophilus parainfluenzae* is a common conditional pathogen in the human respiratory system (48), infections of related N4-like viruses might make them more aggressive. The AMGs of the virulence-associated protein E and the YopX protein were first found in pathogenic *Streptococcus* and *Yersinia* as virulence-associated factors (49, 50) and are encoded by three genomes. Hence, it is hypothesized that these N4-like viruses living in the human gut may be linked to the spread and progress of some bacterial-mediated diseases. In addition, the pre-toxin domain with the VENN motif is only detected in N4-like proviruses, so those were regarded as the N4-like proviral-specific virulence-associated AMGs. This protein domain is found in many bacterial porlymorphic toxins and is located before the C-terminal toxin modules (51). Hence, the related N4-like viral integration might involve the exotoxin-producing cells, giving a competitive advantage to affected bacteria (52).

N4-like viruses were considered a viral lineage with a strictly lytic life strategy (53, 54), but they could also step into a pseudolysogenic stage in some instances (55). Integrase and transposase are essential lysogenic hallmark genes that were used to identify temperate phages (56–58). In addition, the diversity-generating-retroelement (DGR) is also widespread in many temperate bacteriophages and proviruses and is essential for dynamic host adaptations (53–55). Although the genome metadata of Schitoviridae in GenBank labeled eight N4-like proviruses, the lysogenic hallmark genes were not observed in these genomes through functional annotation. Hence, no reliable evidence has demonstrated that N4-like viruses can be involved in the viral lysogenic life cycle. Here, the discovery of 54 genomes of N4-like HQ-UViGs and proviruses that encode lysogenic hallmark genes suggests an N4-like viral lysogenic living strategy (see Table S4 at https://bitbucket.org/minwanglab/n4-like_viruses_supplementary_materials). Four HQ-UViGs encoded both integrase and DGR at adjacent sites in one genomic island, all from the proposed subfamily *SF8*, providing evidence of a lysogeny-specific N4-like viral subfamily with potential high mutagenicity (59, 60). In addition to *SF8*, lysogenic hallmark genes tend to be encoded by specific subfamilies. For the 21 subfamilies encoding these genes, 18 of them have more than 75% of their members encoded lysogenic hallmark genes. The Podoviridae sp. isolate cteks2 (BK028931) is the only member encoding the lysogenic hallmark gene in Humphriesvirinae, indicating that these elements might also undergo horizontal transfer instead of only vertically inheriting in a single lineage. For the hosts infected by these lysogenic N4-like viruses, *Moraxella* is the most common host affected by these viruses. The integrated N4-like proviral regions were detected in 27 *Moraxella* genomic contigs (see Table S4 at https:// bitbucket.org/minwanglab/n4-like_viruses_supplementary_materials). Although diverse proviruses have been revealed that were widespread in the genomes of *Moraxella*, to the best of our knowledge, this is the first report of N4-like viral integration in this bacterial lineage (61). For the habitat origins of these lysogenic N4-like viruses, the marine is the largest habitat that includes 14 lysogenic N4-like HQ-UViGs. They were also detected in various habitats, including freshwater, non-marine saline and alkaline, terrestrial, and

host-associated. This suggests that the biome distribution of these lysogenic N4-like viruses might be much wider than we acknowledged before. Similar to the marine T7-like viruses encoding integrase, the integrase might be obtained from their hosts or other temperate bacteriophages through the recombination process (62). We speculate that the obtaining of lysogenic hallmark genes may provide a selective advantage to N4-like viruses in some nutrient-limited environments with poor host physiology or provide competitive advantages for their host (63–67). The evidence of lysogenic N4-like viruses may increase our knowledge of the living strategies of these viruses.

## Conclusion

Based on the data-mining and filtering protocols, we have identified a set of 342 high-quality N4-like viral genomes. This study has revealed a remarkable expansion of the genome and subfamily diversities of N4-like viruses by over 2.3- and 12.1-fold, respectively. Although N4-like viruses were previously considered to be a viral lineage with a highly conserved core genome (14, 15, 68), our co-phylogenetic and covariance analysis provides evidence of both evolutionary homogeneity and heterogeneity of seven hallmark genes, shedding light on the evolutionary history of N4-like hallmark genes. These viruses can infect not only Proteobacteria but also Firmicutes, which was not previously known. The N4-like viruses are widely distributed in various habitats, including the human gut, oligotrophic open ocean, and polar areas, which were not previously reported. The annotation of the viral pan-proteome revealed the stable and diverse habitat-specific AMGs of N4-like viruses, providing evidence of viral-mediated virulence for some bacteria colonized in host-associated habitats. In addition, the identification of N4-like proviral regions in bacterial genomic contigs and the detection of lysogenic hallmark genes in N4-like viral genomes provided direct evidence of the existence of an N4-like viral lysogenic living strategy. Overall, this study expands our understanding of N4-like viruses on aspects of habitat, genome, evolution, classification, function, and living strategy diversity.

## MATERIALS AND METHODS

### Generation of models to detect N4-like virally associated proteins

The 144 genomes of N4-like viral isolates were retrieved and downloaded from GenBank (2022/8) at the time of writing and were used in reference-guided mining for uncultured N4-like viral genomes in metagenome data sets. We performed a BLAST-based aligning algorithm and an MCL-based clustering method to detect orthologous proteins within viral reference genomes. Notably, 17 N4-like core gene candidates (including nine hallmark gene candidates) were identified in a previous study (23). However, only original genomic sequences were used in this study. All data processing steps were standardized to minimize the error resulting from method differences, including gene calling, protein clustering, data mining, and data filtering (Fig. S7). Hence, some data points (such as generated core gene and hallmark gene candidates) may be inconsistent with previous studies. The gene calling of these reference genomes was performed using Prodigal (v.2.6.3) (69), generating 12,973 viral ORFs. ORFs with less than 50 amino acids (aas) were removed and subjected to Diamond (v.2.0.10) to perform all-to-all BLASTp with the parameters: 1E−10 as the $E$-value, 50% of the minimal number of query covers, 1 as the maximum number of high-scored pairs, and 10,000 as the reporting targets (15, 70). The protein-protein orthologous pairs were screened from the result of all-to-all BLASTp using the criteria of reciprocal-best BLAST (RBB). The protein-protein orthologous pairs were subjected to OrthoMCL (v.2.0.9) to detect N4-like viral protein families (Schitoviridae protein families, SchitoPFs), producing 1,231 and 1,364 N4-like PFs and singletons (SchitoSGs), respectively. Twenty PFs were encoded by nearly all N4-like viruses (encoded by nearly 95% of reference genomes) (core SchitoPFs). The 16 core SchitoPFs were single-copy ORFs in reference genomes; 6 out of the 16 core SchitoPFs were encoded by all N4-like viruses (hallmark genes). The

full-length N4-vRNAP (SchitoPF00019) was not encoded by the genomes of *Gama-leyavirus pSb-1* (GenBank accession: NC_023589) and *Efbeekayvirus vB_AmaP_AD45-P1* (GenBank accession: NC_021532) (9, 71). The N4-vRNAPs of two viruses were recalled as three (gp101-103 on GenBank metadata) and two (metadata mismatched on GenBank) adjacent ORFs, respectively. We speculated that some mistakes might have occurred in the original gene calling of *Efbeekayvirus vB_AmaP_AD45-P1*, resulting in the detection of N4-vRNAP in this virus. In some cases, the stop codons can be ignored in viral ORF transcriptions (read-through) (72, 73). This might be the reason for the N4-vRNAP separation in some rare cases. The separated ORFs from both genomes were concatenated to form the full-length N4-vRNAP. Hence, seven hallmark genes were determined and used to detect and filter the N4-like viral contigs in genome and metagenome databases.

Some N4-like viruses with partial genomes that also encoded hallmark genes were unclassified in GenBank metadata. Hence, an N4-vRNAP-based search against NR (2022/8) was performed first. Multiple sequence alignments (MSA) of the 7 ORFs of hallmark genes were made using the L-INS-I strategy (1,000 iterations) by MAFFT (v.7.490) (74). The seven sets of MSA were used to build specific hidden Markov modules (HMMs) using HMMER (v.3.3.1) (75). The search (1e−10 as the *E*-value) based on seven specific HMMs was performed against the entire viral NR database, producing 971 N4-like viral-associated proteins. In this search, 26 non-complete genomes were classified as Schitoviridae in GenBank, and these encoded at least 3 types of hallmark genes; 10 non-complete genomes were classified as "unclassified Podoviridae," and they also encoded at least 3 types of hallmark genes. The rest of the N4-like viral-associated sequences belong to non-Schitoviridae viral taxonomic units and encode no more than two hallmark genes. Hence, these 10 N4-like viral-associated sequences were determined to be N4-like viruses. The corresponding hallmark genes encoded by the 36 N4-like viruses were added to the ORF set of reference genomes, generating a complete N4-like reference pan-proteome set.

The 154 N4-vRNAP sequences were included in the reference pan-proteome set and were used to build specific HMMs using the same methods as above. The initial search (1e−10 as the *E*-value) based on N4-vRNAP specific-HMM was performed against the entire IMG/VR (v.3) database, producing 243 N4-vRNAP-associated proteins. The 243 sequences from the initial search were combined with the initial 151 N4-vRNAP sequences and used to perform the second iterative HMM-based search using the same methods as above. The result of the N4-vRNAP HMM-based search was converged in 5 iterations, producing 3,786 N4-vRNAP-associated proteins (Fig. S1 and S8A). The UViGs encoding N4-vRNAP-associated proteins were regarded as the N4-vRNAP-associated UViGs and were used to screen reliable N4-like UViGs in the following multistep data-filtering process.

## Filtering of high-quality N4-like viruses

Firstly, considering that the length of N4-vRNAP in reference genomes ranged from 2,842 to 3,820 aas, related proteins with lengths less than 2,500 aas were removed from the results of the fifth HMM-based search; this resulted in 1,984 target proteins (Fig. S4A). In this step, nearly all T7-like RNA polymerases that had a distant relationship with N4-vRNAP were removed. Secondly, a conserved motif ("ExDGxxxG"), which was located between 4,563 and 4,570 aas in the MSA, was determined from the N4-vRNAP in the reference genomes (Fig. S5). All 1,984 target proteins were screened by this conserved motif, producing 983 target proteins. Thirdly, some data were not permitted to be used in accordance with the data utilization policy of JGI; hence, all related data with restricted utilization were removed from the 983 target proteins, producing 859 target proteins. Fourthly, to remove potential duplications genome-wide, the corresponding UViG genomic sequences of the 859 target proteins were subjected to CD-Hit (v.4.7) (76) to remove the duplications with the 99% similarity threshold under global mode, producing 601 non-redundant UViGs.

The final filtering step was based on the number of types and copies of the other six hallmark genes encoded by 601 viral genomes. Other pre-built specific HMMs of six hallmark genes were subjected to HMMER (v.3.3.1) to perform an HMM-based search against predicted ORFs of the 601 non-redundant UViGs (75). These iterative searches were performed with the same methods mentioned above, producing 475, 442, 436, 395, 410, and 401 target proteins of N4-vDNAP, N4-MCP, N4-Portal, N4-Tail, N4-TerL, and N4-TerS, respectively (Fig. S1). The corresponding genomes encoding all seven hallmark genes were regarded as high-quality N4-like viral genomes, including 148 genomes/UViGs from GenBank and 148 HQ-UViGs from IMG/VR (v.3). The completeness of all viral genomes was determined through CheckV (see Table S1 at https://bitbucket.org/min-wanglab/n4-like_viruses_supplementary_materials) (77).

## N4-like proviral region determined from prokaryotic whole genome sequencing

No proviruses were found in IMG/VR (v.3) based on the provided metadata. To survey the endogenization of N4-like viruses in prokaryotic genomes, we perform N4-like viral genomic signature detection based on seven hallmark genes. First, the N4-vRNAP-HMM was built based on 302 genomes generated in the last step and was used to search potential N4-vRNAP homologs in the cellular NR (2022/08) database with the same parameters used in the above step. This step generated 8,991 N4-vRNAP homologs, which were harbored in 460 prokaryotic genomic assemblages (see Table S5 at https://bitbucket.org/minwanglab/n4-like_viruses_supplementary_materials; Fig. S1). Second, the related assemblages were retrieved and downloaded from the whole genome sequencing database maintained by NCBI. As some N4-like viruses have genomic lengths over 100 kbp and to overcome some viral element contamination in bacterial genome sequencing (such as virocell, this viral element contamination might be regarded as part of some bacterial genome, introducing erroneous metadata and artifacts), only bacterial genomes with lengths over 100 kbp were retained in the following filtering step. Other six N4-like viral hallmark genes were used to filter these assemblages, as mentioned above, resulting in 40 prokaryotic genomic contigs. Third, the 40 contigs were subjected to CheckV (v. 0.8) to determine proviral regions (see Table S1 at https://bitbucket.org/minwanglab/n4-like_viruses_supplementary_materials). The N4-like proviral regions were extracted and added to the N4-like viral genome data set and labeled "Provirus" in our metadata.

## Host prediction of uncultured N4-like viruses

The host association of viruses collected by IMG/VR was performed based on Earth's Virome protocol (EVP), established by Paez-Espino and colleagues (19, 20) and was continuously developed in the construction of IMG/VR (v.3), including four main approaches (CRISPR-spacer matching, viral-host tRNA gene matching, viral-host genome-content similarity, and consensus virally operated taxonomic units host) (25). As a supplementary, the updated CRISPR-spacer database (released in April 2022) maintained by CRISPR-Cass++ (24, 78, 79) was downloaded and used to perform CRISPR-spacer matching against high-quality N4-like viral genomes generated by this study using the same parameters with EVP: 100 as the percentage of identity and spacer sequence coverage. This produced nine additional viral-host associations, which were merged with host information provided by IMG/VR (v.3) as the final N4-like viral host metadata (see Table S1 at https://bitbucket.org/minwanglab/n4-like_viruses_supplementary_materials).

## Phylogenic trees of N4-like viruses

The N4-like viral lineage phylogenic tree was calculated based on the concatenated ORFs of seven hallmark genes. These genes were subjected to MAFFT (v7.490) to make MSA (with the L-INS-I strategy and 1,000 iterations) (74). The concatenated MSA of the seven ORFs of hallmark genes was subjected to iqtree2 to calculate the maximum-likelihood

phylogenic tree with 1,000 bootstraps (80), with LG + F + R9 as the suggested substitution module. The single protein trees of seven hallmark genes were calculated using a similar method, with Q.pfam + F + R9, LG + F + R7, LG + F + R9, Q.pfam + F + R7, Q.yeast + R9, Q.yeast + F + R7, and Q.pfam +F + R10 as the suggested substitution modules for N4-vDNAP, N4-MCP, N4-Portal, N4-Tail, N4-TerL, N4-TerS, and N4-vRNAP, respectively. The newick files of the maximum-likelihood trees were visualized and annotated by iToL (v.5) (81).

The global phylogenetic tree of N4-like is calculated based on the bias of N4-TerL and related protein homologs encoded by other viruses. The HMM of 342 N4-TerL protein sequences was compared against the NCBI viral RefSeq (2022/08) database to search protein homologs with the same parameters mentioned above, resulting in 133 non-Schitoviral N4-TerL homologs. All 475 proteins were aligned and subjected to iqtree2 to calculate the maximum-likelihood phylogenic tree with the same parameters mentioned above (80). The newick files of the maximum-likelihood trees were visualized and annotated by iToL (v.5) (81).

The co-phylogeny between N4-like viral hallmark genes was characterized by the sRFVs calculated by TreeKO between two transient phylogenic trees (82). The sRFVs are used to describe how many leaf splits are implied by only one of the compared trees (83). The co-phylogenic trees of identified high-quality N4-like viruses were pruned from the original trees and visualized by an R script.

## Covariance of N4-like viral hallmark genes

The interprotein COrrelated Mutations Server was used to assess the covariance between N4-like viral hallmark genes at the amino acid level (84). The protein sequences of the N4-like viral hallmark gene were aligned using MAFFT (v7.490) (74) with the same parameters mentioned above and submitted to the server as 21 transient protein pairs. A concatenated MSA for each protein pair from the same genome was used to obtain the quantitative covariance characterization. The covariance was described as an mfDCA value, which is one of the state-of-the-art co-evolution algorithms (85, 86). The mfDCA is independent of the phylogenetic history and thus provides a more reliable prediction of the interacting residues between protein partners (26). The top 1% of covariance scores between protein pairs were selected as highly interacting residues (see Table S2 at https://bitbucket.org/minwanglab/n4-like_viruses_supplementary_materials) and were visualized by an R script (Fig. 6). Based on the previous study, the protein sequences of integrase and terminase from *Lambdavirus Lambda* were used as the negative control; the protein sequences of portal protein and terminase from *Lambdavirus Lambda* were used as the positive control (26). Protein pairs were considered to have high covariance if their medians of mfDCA values were higher than the medians of positive and negative controls. The differences in significance between protein pairs and controls were checked through a *t*-test.

## The pangenome characterization and classification of N4-like viruses

The genome-content-based network was used to detect genomic relationships among N4-like viruses. The ORFs of the 342 high-quality N4-like viral genomes with a length of more than 50 aas were subjected to Diamond (v2.0.10) to perform the all-to-all BLASTp with the same parameters as for RBB criteria (70), producing 1,383,722 protein-protein orthologous pairs. These protein-protein orthologous pairs were subjected to vConTACT2 (87) to detect protein families through the MCL algorithm, producing 48,021 SchitoPFs. These SchitoPFs were used to cluster N4-like viral genomes through ClusterONE by vConTACT2, producing 29 N4-like VCs. The genomic relationship weights produced by vConTACT2 were subjected to Gephi (v.0.9) (88) to calculate layout with the Force Atlas algorithm.

To characterize the pangenome feature of the N4-like viruses, SchitoPFs and SchitoSGs generated through MCL (two as the inflation) were subjected to vegan (v.2.5)

(89) to generate PF accumulation curves (with 300 randomly sampling per accumulation) for N4-like viruses from GenBank and IMG/VR (v.3), respectively.

To assign the taxonomic units of the N4-like viruses following the criteria of ICTV (13, 23), VCs generated by vConTACT2 and intergenomic similarity clustering by VIRIDIC were used to assign the N4-like viral subfamily/genus-level groups (90). All proposed taxonomic units of N4-like viruses were manually inspected to meet the criteria of the ICTV proposal on Schitoviridae. The VCs generated by vConTACT2 were mapped on the average nucleotide identity (ANI) heatmap generated by VIRIDIC to determine the boundary of each subfamily. We found that 20% ANI seems to be the most suitable threshold to distinguish eight established subfamilies, while the Enquatroviri- nae/Rothmandenesvirinae and Fuhrmanvirinae/Pontosvirinae were merged under this criterion. The Fuhrmanvirinae and Pontosvirinae were grouped as a single VC by vConTACT2, implying that they have a much closer genome-content relationship than other subfamilies. However, the Pontosvirinae were separated into multiple subfamilies if setting a higher ANI threshold. This indicates that the ANI threshold of subfamilies within Schitoviridae still lacks a unified standard value. Hence, our proposed subfamilies shall be a minimal conservative estimate result, which may include more subfamilies but shall not be less than this number ($n$ = 89) (see Table S1 at https://bitbucket.org/minwanglab/n4- like_viruses_supplementary_materials). The genus/species-level units of high-quality N4-like viruses were auto-classified by VIRIDIC with the all-to-all BLASTn parameters: seven as the word size, two as the reward, three as the penalty, five as the gap open, and two as the gap extend. The result of all-to-all BLASTn was used to cluster the criteria: 95% as the species similarity threshold and 70% as the genus similarity threshold. The similarity was normalized by the total length of each genome.

## Functional annotation of N4-like viral proteome

The pan-proteome of high-quality N4-like viruses was subjected to kofam and pfam_scan.pl to perform an HMM-based search (1e−5 as $E$-value) against the KO database (2022-1) (91) and the Pfam-A (v.35) database (92), respectively. The 1,101 ORFs that were probably involved in metabolic host reprogramming were identified. The results from the Pfam-A annotation were manually assigned to a suitable pathway defined by the KEGG metabolic pathway database. All determined AMGs were mapped into the KEGG metabolic pathways, belonging to 18 main metabolic categories and 90 functional types.

## ACKNOWLEDGMENTS

This work was partly supported by the Laoshan Laboratory (No. LSKJ202203201), the National Key Research and Development Program of China (No. 2018YFC1406704, 2022YFC2807500), the Natural Science Foundation of China (No. 41976117, 42120104006, 42176111, and 42188102), the Fundamental Research Funds for the Central Universities (No. 202172002, 201812002, 202072001, and Andrew McMinn), the Marine S&T Fund of Shandong Province (No. 2018SDKJ0406-6), the Key R&D Project in Shandong University (No. 2019GHY112079), and the MEL Visiting Fellowship Program (No. MELRS1511).

K.Z., Y.L., and M.W. provided the concept for this work; D.P.-E. curated the data; K.Z. and Y.L. performed the formal analysis; X.Z., C.G., and H.S. provided the background investigation; K.Z. designed the method and approaches; Y.Y.S., W.J.M., L.L.W., and Y.-Z.Z. administrated this project; D.P.-E. provided the data resource; K.Z., J.T., and F.C. provided the advice on software; K.Z., N.J., C.A.S., and J.H. performed the result visualization; A.M. validated this work; K.Z. wrote the original manuscript; Y.L. and A.M. reviewed the manuscript; Y.L., A.M., and M.W. supervised this project and provided funding.

## AUTHOR AFFILIATIONS

[1]Key Laboratory of Polar Oceanography and Global Ocean Change, Frontiers Science Center for Deep Ocean Multispheres and Earth System, College of Marine Life Sciences, Institute of Evolution and Marine Biodiversity, Ocean University of China, Qingdao, China

[2]UMT-OUC Joint Centre for Marine Studies, Qingdao, China

[3]DOE Joint Genome Institute, Lawrence Berkeley National Laboratory, Berkeley, California, USA

[4]Mammoth Biosciences Inc., South San Francisco, California, USA

[5]Qingdao Central Hospital, Qingdao, China

[6]Institute of Marine Biotechnology, Universiti Malaysia Terengganu (UMT), Kuala Terengganu, Malaysia

[7]State Key Laboratory of Microbial Technology, Marine Biotechnology Research Center, Shandong University, Qingdao, China

[8]Key Laboratory of Physical Oceanography, Ministry of Education, Ocean University of China, Qingdao, China

[9]Institute of Marine and Environmental Technology, University of Maryland Center for Environmental Science, Baltimore, Maryland, USA

[10]State Key Laboratory of Marine Environmental Sciences, Institute of Marine Microbes and Ecospheres, Xiamen University, Xiamen, China

[11]Department of Earth, Ocean and Atmospheric Sciences, Institute for the Oceans and Fisheries, The University of British Columbia, Vancouver, British Columbia, Canada

[12]Department of Microbiology and Immunology, Institute for the Oceans and Fisheries, The University of British Columbia, Vancouver, British Columbia, Canada

[13]Department of Botany, Institute for the Oceans and Fisheries, The University of British Columbia, Vancouver, British Columbia, Canada

[14]SOA Key Laboratory for Polar Science, Polar Research Institute of China, Shanghai, China

[15]Institute for Marine and Antarctic Studies, University of Tasmania, Hobart, Tasmania, Australia

[16]The Affiliated Hospital of Qingdao University, Qingdao, China

## AUTHOR ORCIDs

Yantao Liang  http://orcid.org/0000-0002-2477-8197
Feng Chen  http://orcid.org/0000-0002-6064-7805
Curtis A. Suttle  http://orcid.org/0000-0002-0372-0033
Andrew McMinn  http://orcid.org/0000-0002-2133-3854

## FUNDING

| Funder | Grant(s) | Author(s) |
|---|---|---|
| Marine S&T Fund of Shandong Province | 2018SDKJ0406-6 | Min Wang |
| MOST \| National Key Research and Development Program of China (NKPs) | 2018YFC1406704 | Min Wang |
| MOST \| National Natural Science Foundation of China (NSFC) | 42188102 | Min Wang |
| Central University Basic Research Fund of China (中央高校基本科研专项) | 201812002 | Andrew McMinn |
| MOST \| National Natural Science Foundation of China (NSFC) | 41976117 | Min Wang |
| MOST \| National Natural Science Foundation of China (NSFC) | 41606153 | Yantao Liang |
| Key R&D Project in Shandong University | 2019GHY112079 | Min Wang |
| MEL Visiting Fellowship Program | MELRS1511 | Min Wang |
| Laoshan Laboratory | LSKJ202203201 | Min Wang |

| Funder | Grant(s) | Author(s) |
|---|---|---|
| MOST \| National Key Research and Development Program of China (NKPs) | 2022YFC2807500 | Min Wang |
| MOST \| National Natural Science Foundation of China (NSFC) | 42120104006 | Min Wang |
| Central University Basic Research Fund of China (中央高校基本科研专项) | 202072001, 202172002 | Min Wang |
| Central University Basic Research Fund of China (中央高校基本科研专项) | | Andrew McMinn |

## AUTHOR CONTRIBUTIONS

Kaiyang Zheng, Conceptualization, Formal analysis, Methodology, Software, Visualization, Writing – original draft | Yantao Liang, Conceptualization, Formal analysis, Funding acquisition, Writing – review and editing | David Paez-Espino, Data curation, Resources | Xiao Zou, Investigation | Chen Gao, Investigation | Hongbing Shao, Investigation | Yeong Yik Sung, Project administration | Wen Jye Mok, Project administration | Li Lian Wong, Project administration | Yu-Zhong Zhang, Project administration | Jiwei Tian, Software | Feng Chen, Software | Nianzhi Jiao, Visualization | Curtis A. Suttle, Visualization | Jianfeng He, Visualization | Andrew McMinn, Funding acquisition, Validation, Writing – review and editing | Min Wang, Conceptualization, Funding acquisition, Supervision

## DATA AVAILABILITY

Viral contigs reported in this study are available on GenBank (https://www.ncbi.nlm.nih.gov/nuccore/?term=) or IMG/VR (https://img.jgi.doe.gov/cgi-bin/vr/main.cgi), whose accessions are shown in Table S1 (https://bitbucket.org/minwanglab/n4-like_viruses_supplementary_materials). Tables S1 to S5, all FASTA-formatted protein families, and newick-formatted files of phylogenetic trees were on the repository host site (provided at https://bitbucket.org/minwanglab/n4-like_viruses_supplementary_materials).

## ADDITIONAL FILES

The following material is available online.

### Supplemental Material

**Fig. S1 (mSystems00197-23-s0001.pdf).** The genome organization of N4-like viruses.
**Fig. S2 (mSystems00197-23-s0002.pdf).** Identification pipeline for N4-like viruses.
**Supplemental text (mSystems00197-23-s0003.docx).** Legends for supplemental figures.

### Open Peer Review

**PEER REVIEW HISTORY (review-history.pdf).** An accounting of the reviewer comments and feedback.

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
