## [Reviewer comments · mSystems]

Identification of hidden N4-like viruses and their interactions with hosts

Kaiyang Zheng, Yantao Liang, Antonio David Paez-Espino, Xiao Zou, Chen Gao, Hongbing Shao, Yeong Sung, Wen Mok, Li Wong, Yu-Zhong Zhang, Jiwei Tian, Feng Chen, Nianzhi Jiao, Curtis Suttle, Jianfeng He, Andrew McMinn, and Min Wang

Corresponding Author(s): Yantao Liang, Ocean University of China College of Marine Life Sciences

Review Timeline:

Submission Date:	March 6, 2023
Editorial Decision:	June 1, 2023
Revision Received:	July 19, 2023
Accepted:	July 19, 2023

Editor: Jack Gilbert

Reviewer(s): The reviewers have opted to remain anonymous.

Transaction Report:

DOI: <https://doi.org/10.1128/msystems.00197-23>

June 1, 2023

Prof. Yantao Liang
Ocean University of China College of Marine Life Sciences
No. 5 Yushan Road,
Qingdao, Shandong 266000
China

Re: mSystems00197-23 (**Identification of hidden N4-like viruses and their interactions with hosts**)

Dear Prof. Yantao Liang:

Thank you for submitting your manuscript to mSystems. We have completed our review and I am pleased to inform you that, in principle, we expect to accept it for publication in mSystems. However, acceptance will not be final until you have adequately addressed the reviewer comments.

Preparing Revision Guidelines

Please return the manuscript within 60 days; if you cannot complete the modification within this time period, please contact me. If you do not wish to modify the manuscript and prefer to submit it to another journal, please notify me of your decision immediately so that the manuscript may be formally withdrawn from consideration by mSystems.

Sincerely,

Jack Gilbert

Editor, mSystems

Journals Department

Reviewer comments:

Reviewer #1 (Comments for the Author):

#Summary. The authors have made many updates in this revision, including efforts to address the major comments previously raised regarding removing the HGT analysis, incorporating a search for prophages in bacterial genome assemblies, and adding a CRISPR-spacer based analysis to predict hosts. The authors have also improved figures and added helpful new figures and provided useful data for the reader in a public github repository. These revisions have strengthened the manuscript and added value for the readers and it is great to see this diversity of N4-like phages and to learn about their host associations. However, I still have major concerns, including around issues from the initial review, these are highlighted below. The comments are long, but I sincerely hope they will be viewed as helpful by the authors.

#Major Comments

##Prophage analyses - missing cases: Exploration of the author's results suggests that there are N4-like phages in GenBank that have been missed in this re-analysis (e.g. *Acinetobacter* contigs APOR01000031.1 and CP110465.1 and JAIGUQ010000009, all of which appear to contain all 7 N4-like hallmark genes based on BLAST searches, and more cases).

- All of the aforementioned examples represent bacterial contigs that are <100kb, the cut-off used to define which contigs to evaluate in searching the database of bacterial assemblies (L435-L438). Though the intent to be conservative and not introduce false positives is good, this cut-off is problematic because it precludes discovery of phages that exist as extrachromosomal elements (as plasmids), and because assembly-artifacts (e.g. coverage or repeats) can lead to integrated prophages being assembled as distinct contigs. Given that the N4-likes have an average genome size of ~70kb, the 100kb cut-off would be expected to result in such losses.
- Further, though the first 2 examples mentioned above have nearly identical vRNAP amino acid sequences, only one of the pair (APOR01000031.1) is listed in the initial Table S5 vRNAP-based HMM-search against GenBank, and it is not clear why the other should not have also been detected, this raises the question of whether additional such cases are also missing.
- Also, one of these N4-likes (CP110465) is described in GenBank as a plasmid (pRBH2-3), and there are many additional apparently N4-like *Acinetobacter baumannii* phages that may be plasmids. If this is the case then this would be an interesting additional life-history strategy for this group that may not have previously been described (?) and would thus be worth deeper exploration for other examples. Many of these missing N4-likes are also from bacterial strains from a published study focused on carbapenem resistant *Acinetobacter baumannii* (CRABS, <https://doi.org/10.1016/j.csbj.2021.12.038>), which are a medically important group. Altogether, this finding, together with the finding of integrated versions, suggests the possibility that their use in phage therapeutics should be carefully considered.
- Searching with proteins from these examples against both IMG/VR or UniProtKB (with jackhmmer) also identifies additional cases of likely N4-likes that are not represented in Tables S1 or S5. For example, searching with the TerL from CP110465 (UZG64161.1) against UniProtKB (all sequences also in GenBank) identifies an *Alteromonas* contig CP031010.1 (among others) that also contains a nearby vRNAP. This vRNAP is not in Table S5 and hits predominantly to viruses in the Schitoviridae. This suggests that additional N4-like diversity that would be hit by the 7-hallmarks identified by the authors is being missed.
- Finally, it seems that the prophages in IMG/VR (UViG source = isolate) were not included in the analyses along with the UViGs; if the authors plan to repeat or expand any analyses it would likely be beneficial to take advantage of the updated IMG/VR v4 with the geNomad phage calls.
- An exploratory look shows that many phage sequences identified by the authors here as N4-like currently have no family designation in IMG/VRv4 - this highlights the contribution and value of the author's work.

##Identification and characterization of AMGs: Numerous AMGs are identified in this work, and the authors have now grouped these as Class I or II, as suggested. However, it is not clear that these have been thoroughly curated to ensure they do not represent phage-adjacent bacterial genes, or simply phage genes involved primarily in processes directly relating to phage replication rather than shaping of host metabolism. For example, in the DRAM-v manuscript it is highlighted that, in identification of AMGs, "DRAM-v also flags users to the probability of a gene being involved in viral benefit rather than enhancing host metabolic function (e.g. certain peptidases and CAZymes are used for viral host cell entry (Figure 6B))." <https://doi.org/10.1093/nar/gkaa621>. Though CheckV was applied it was not mentioned whether VirSorter2 or another phage finder was used in addition (prior to CheckV), which would likely be necessary in the case of >100kb bacterial contigs. If this is added in subsequent analyses I suggest considering using geNomad (<https://github.com/apcamargo/genomad>).

#Additional Comments

##Prophage analyses - likely Lactobacillales gram-positive contaminant: As mentioned in the initial review, the finding of an N4-like prophage in a gram-positive is highly unexpected (it being the only case of a potential gram-positive bacterial host). Unless additional rigorous steps are taken to rule out the potential for contamination, for example requesting and growing the strain and inducing out the predicted prophage, it is important that the authors be far more cautious in their representation and lean

towards representing this as a likely contaminant. Given the high identity of the hits to *E. faecalis* N4-phages this likely represents a contamination and assembly artifact.

##Host prediction: Provide a statement in the methods about the number of additional host predictions achieved using the additional CRISPR search.

##Category names: The analyses throughout refer to a category of "Provirus" and a category of "HQ-UViG" (e.g. Fig. 3C, Table 1), on the basis of the pipeline used to identify them, however, this can be confusing as IMG/VR also includes prophages identified in bacterial genomes (from GenBank), not just metagenomes. Thus IMG/VR hits would be expected to also have sequences in the "Provirus" category and some overlap with hits identified by the direct searches in GenBank. In addition, in some places the viruses identified in the second round GenBank search are also referred to as UViGs (in cases where the number 158 is used, e.g. L117), which seems is otherwise reserved for use for the IMG/VR pipeline. Also, it is difficult to follow the #s as sometimes the sets referred to are 144, 10, 40, 148; other times 158 is referenced, other times 611, other times 154.

##Clarity: The manuscript would benefit from review of spelling, grammar, and clarity of phrasing.

#Figures

##Fig. 1 It is not clear what is meant by "repartition"

##Fig. 2 Figure Legend: The legend states (L847) that the outer ring is the ecosystem for the HQ UViG but some leaves that are identified by red-colored UViG branches do not have corresponding ecosystem assignments, likewise, in the second ring in from the outside the legend describes this (L848) as referring to experimental host lineages of N4-like viral isolates but leaves identified as UViGs (red branches) also have colorings, and the legend states (L850) that there is a ring "V" but this is not shown. "Associated genomes" should probably be "contigs" for L851? Figure Panel A: In the legend in the image the line for "provirus" is missing. Figure Panel B: A number of aspects of this figure would benefit from clarification. Please provide the total number of contigs represented by each category of 1-7 cores and provide further interpretation of the figure in the legend or body. For example, it is unclear why there is not a consistent decrease in the number of contigs with increasing number of cores in each ecosystem, in other words, one would expect that a more relaxed criterion (1 core) would yield a greater number of hits than a more strict criterion (7 cores). Also, it is not clear why the average contig size for each ecosystem (e.g. ~100kbp for Aquatic and apparently considering all #s of cores) appears to be larger than the maximum for the 7 cores across ecosystems (~70kbp). It is not clear what datasets were used or what the interpretation of these calculations is. L135-L140 It is also not clear here which genomes are being used - the authors refer to HQ-UViGs but then state that the marked difference in sizes are due to levels of completeness in genomic assemblies, if only contigs with all 7 cores are being considered, then this raises the question of whether longer assemblies reflect poor trimming of host sequences (with potential downstream consequences for AMG analyses). This point requires clarification.

##Fig. 3 Nice addition and cross-referencing, very informative

##Fig. 4 Figure legend: What are "floated genera" (L881)? If this refers to other genera that are currently pending as taxonomy proposals please provide a reference. Panel B: What do the yellow and green colorings by the leaves indicate? The legend labeled "Branch" intermingles branch line style with individual leaf coloring. Please name the outgroup.

##Fig. 5 In analyses relating to the multigene phylogeny in relation to other gene phylogenies, was gene-length controlled for (to prevent longer genes from dominating the signal)? Panel B/C: The quantitative basis for defining the red boxes is not clear and I suggest omitting them; clarify that the "Top ten subfamilies" are the top ten most abundant (presumably?); however, it is not clear why all subfamilies would not have been labeled.

##Fig. 6 Though the idea of including a figure to facilitate interpretation is a nice one, it is not clear how to interpret the cartoon. For example, the arrows from the label for portal point to multiple other points in the cartoon but it is not clear what they are pointing to or why (e.g. are all of the circles along the tail meant to represent the major tail protein? If so, they should all be the same size and all arrows should point to one exemplar not to different ones). The plot itself is also not well described, please provide description of what all the points represent, how many there are for each comparison, what their colors mean and whether they relate to the cartoon and if so how, and what the varying color intensity is meant to represent. The large and small subunit terminase appear to be shown as part of the virion - however, this is incorrect as they are not incorporated into the virion but are involved in packaging (e.g. https://viralzone.expasy.org/3944?outline=all_by_species).

##Fig. 7 Piecharts: The slice for "Others" is colored green as a Class I AMG, however the expanded view shows that it contains Class I and Class II AMGs, if this is the case then a third color should be used for "other"; also "Cell circle" should be "Cell cycle", here and throughout (e.g. Table 1, L265, L266). Cell cartoon: The intent to map AMGs to the cell to highlight processes is nice, however it is a bit challenging to interpret some aspects of this figure as there is a lot of information and not all the components are labeled and for some of the labels it is not known whether the cartoon represents an example of a possible function (e.g. Nutritional limitation SF64), streamlining some pieces here would be helpful.

##Fig. S1-S6 - Recommend combining all of these genome diagrams into a single continuous multi-page figure that includes all 342 identified viruses all ordered onto a tree (perhaps the concatenated hallmark tree); this will only be readily viewable to the reader by zooming in quite a bit and scrolling up and down, but for anyone interested in this level of detail they will appreciate having everything on one page. Also, recommend re-coloring the genes such that all 7 core genes have a highly saturated bold color as well as the bold outline (rather than all being white and only marked by an outline) as these provide common anchor points to facilitate rapid visual qualitative assessment and comparison of the genomes. Additional genes of interest to the author could still be colored but perhaps in less saturated colors, for example. Also, it would be visually helpful to include the genome backbone.

##Fig. S7 - It is very helpful and nice to have this overview figure. Some additional points would benefit from clarification:

- Legend: The legend the numbers don't match the figure, e.g. the legend refers to 158 high quality UViGs including the 148 from IMG/VR and the 10 from GenBank, however there is no reference to 158 in the figure and the 10 from GenBank are not

identified as HQ-UViGs in the figure so it is not clear for the reader.

- Figure: I suggest adding headers on the top of the left and right sides & calling them panels A & B.
 - Panel A: Suggest removing the line from "144 N4-like viral isolates" to "10 additional N4-like ...".
 - Panel A: Suggest changing: "length of N4-vRNAP" to "length of N4-vRNAP (req 2500aa min)", "N4-associated UViGs" to "N4-like-vRNAP-candidate UViGs" (at least 4 places); "CD-HIT: 99%..." to "remove near-identical sequences - CD-HiT...".
 - Panel A: The last section leading from the 601 to the 302 is not clear. There is no process box (green box) connecting the 601 to the 148 to explain how this number was reduced. It seems the bottom left section is meant to show the process reflecting searches with HMMs from all the hallmark genes to reduce the number from 601 to 148, if this is the case it would clearer to remove all of those boxes which are redundant with the top left of the figure and simply indicate this as a green box between the 601 and 148 with a statement with something like "search with 7 hallmark gene HMMS from 154 N4-like viral genomes from GenBank".
 - Panel A: I suggest coloring the boxes for all the unique sets with some new color -> "144 N4-likes", "10 additional ...", "148 HQ-UViGs", and "40 N4-like", as these are the final ones being added up.
 - In panel B: Suggest removing "free-living" from "Free-living N4 ..." to "N4-vRNAP hallmark-HMM"
 - In panel B: Why is this difference between the # of hits (8991) and the number of assemblages (460) so great? Are there many copies commonly found in each of the assemblages? Why would this be?
 - In panel B: Suggest changing "co-occurrence of all six hallmark genes" to "co-occurrence of all 7 hallmark genes"
- ##Fig. S8 This figure is not referenced in the manuscript and it is not clear what panel B is showing, for example why are there different numbers of iterations of HMM and how does this inform the reduction of the 601 HQ-UViGs to the 148?

Dear reviewer,

Many thanks for your professional comments which are much beneficial to improve this work. Based on your suggestion and request, we have made second corrected modification of on the revised manuscript. Some questions you have raised are highly insightful, but unfortunately, we have not been able to fully resolve them at this time. However, we still believe that this work can make some contributions to this field. We would like to show the details as follows:

Major comments

1. Exploration of the author's results suggests that there are N4-like phages in GenBank that have been missed in this re-analysis. The aforementioned examples represent bacterial contigs that are <100kb. Given that the N4-likes have an average genome size of ~70kb, the 100kb cut-off would be expected to result in such losses.

Response: We acknowledge that our results may have missed some positive findings, but we chose a 100-kbp cutoff to screen for endogenous N4-like viruses based on the identification of nine N4-like viral genomes exceeding 100 kbp in our dataset, including one isolate (*Alteromonas* phage vB_AmaP_AD45-P1). Therefore, we believe that a cutoff lower than 100 kbp would introduce potential contamination from exogenous viral fragments. As a result, we accepted the loss of some positive results to ensure the reliability of the dataset. We have added corresponding descriptions in the Methods section. (The “Methods”: “N4-like proviral region determine from prokaryotic whole-genome-sequences (WGS)”).

2. Only one of the pair (APOR01000031.1) is listed in the initial Table S5 vRNAP-based HMM-search against GenBank, and it is not clear why the other should not have also been detected, this raises the question of whether additional such cases are also missing.

Response: We sincerely appreciate the thorough review by the reviewers regarding the data generated in this study. The differences in versions of data mining software and databases may have resulted in the loss of some results. However, we believe that this loss represents only a small fraction, and our current results still adequately represent the diversity of endogenous N4-like viruses. In practice, maintaining up-to-date data is

challenging. This study began in early 2022, but we intend to continue updating the diversity of N4-like viruses, especially based on the now updated IMG/VR (v.4) database. With our current results, we believe that we have effectively expanded the understanding of the diversity of N4-like viruses in multiple dimensions, providing valuable research material for other researchers in the field.

3. One of these N4-likes (CP110465) is described in GenBank as a plasmid (pRBH2-3), and there are many additional apparently N4-like *Acinetobacter baumannii* phages that may be plasmids. If this is the case then this would be an interesting additional life-history strategy for this group that may not have previously been described ?

Response: Previously, it has been reported that N4-like viruses can enter a pseudolysogenic stage in the form of plasmids (10.1371/journal.pone.0215456). Therefore, from this perspective, N4-like viruses cannot be strictly classified as virulent phages. This aspect has been described in the section titled "Host metabolic reprogramming potential and putative lysogenic features of N4-like viruses." (L299).

4. Searching with proteins from these examples against both IMG/VR or UniProtKB (with jackhmmer) also identifies additional cases of likely N4-likes that are not represented in Tables S1 or S5.

Response: An explanation has been provided in the aforementioned second point.

5. It seems that the prophages in IMG/VR (UViG source = isolate) were not included in the analyses along with the UViGs; if the authors plan to repeat or expand any analyses it would likely be beneficial to take advantage of the updated IMG/VR v4 with the geNomad phage calls.

Response: IMG/VR (v.3) has previously annotated a subset of viruses derived from cellular genomes as "proviruses." We have carefully checked the data sources for this study and indeed found no UViGs labeled as "proviruses." Therefore, the N4-like proviruses provided in this paper are newly analyzed results added during the first revision process. As mentioned by the reviewers, we intend to continuously update the diversity of N4-like viruses based on IMG/VR (v.4). However, for the research efficiency, we do not plan to include a complete reanalysis of the entire project based on IMG/VR (v.4) in this manuscript.

6. It is not clear that these have been thoroughly curated to ensure they do not represent phage-adjacent bacterial genes, or simply phage genes involved primarily in processes directly relating to phage replication rather than shaping of host metabolism.

Response: Regarding the concerns raised by the reviewers regarding the data in this manuscript, we would like to address them as follows:

1) For UViGs, all contigs are derived from IMG/VR (v.3) rather than the original assembly contigs. This dataset was generated using the Earth Virome Pipeline (EVP) designed by David et al., which includes screening and quality assessment of viral contigs. As a downstream analysis, we did not perform further detailed cleaning of these contigs.

2) For proviruses, we took a series of steps to reduce contamination from cellular fragments (although it inevitably resulted in the loss of some positive results). The provided endogenous viral fragment loci do not extend beyond the boundaries (both ends) of N4-like viral hallmark genes.

3) Methodologically, we followed the approach of mining NCLDV data (10.1038/s41586-020-1957-x) and filtered the sequences based on seven conserved N4-like viral hallmark genes. We provided both high-quality genomes (encoding all seven core genes) and low-quality genomes (encoding less than seven core genes) for subsequent researchers to reproduce most of our results.

The results of this study also demonstrate a close correlation between the newly discovered N4-like viruses and reference genomes, both in terms of phylogenetic relationships and genome-content relationships. We acknowledge the potential contamination from cellular organisms, but horizontal gene transfer between viruses and cellular organisms is known to be frequent, and there is currently no perfect method to completely avoid contamination from cellular organisms in viral datasets. In conclusion, we believe that the data generated in this study are sufficiently rigorous to provide foundational research material for subsequent researchers.

7. Provide a statement in the methods about the number of additional host predictions achieved using the additional CRISPR search.

Response: Through CRISPR-spacer matching, we have identified 8 new viral-host associations. The relevant description has been added to the Methods section under "Host

prediction of uncultured N4-like viruses" (L445).

8. The analyses throughout refer to a category of "Provirus" and a category of "HQ-UViG" (e.g. Fig. 3C, Table 1), on the basis of the pipeline used to identify them, however, this can be confusing as IMG/VR also includes prophages identified in bacterial genomes (from GenBank), not just metagenomes.

Response: An explanation has been provided in the aforementioned 5th point. We have carefully checked the data sources for this study and indeed found no UViGs labeled as "proviruses." We have added related description in "Methods" ("N4-like proviral region determine from prokaryotic whole-genome-sequences (WGS)") (L420).

Additional comments

9. Fig. 1 It is not clear what is meant by "repartition"

Response: The related sentence was deleted to avoid ambiguity (Legend of Fig. 1).

10. Fig. 2 Figure Legend: The legend states (L847) that the outer ring is the ecosystem for the HQ UViG but some leaves that are identified by red-colored UViG branches do not have corresponding ecosystem assignments, likewise, in the second ring in from the outside the legend describes this (L848) as referring to experimental host lineages of N4-like viral isolates but leaves identified as UViGs (red branches) also have colorings, and the legend states (L850) that there is a ring "V" but this is not shown. "Associated genomes" should probably be "contigs" for L851?

Response: I apologize for our oversight. We have revised the figure legend to accurately describe the information in the image. We have changed "Experimental host lineages" to "Experimental or predicted host lineages."

11. Fig. 2 Panel A: In the legend in the image the line for "provirus" is missing.

Response: I apologize for our oversight. The legend of "proviruses" has been added.

12. Fig 2 Panel B: A number of aspects of this figure would benefit from clarification. Please provide the total number of contigs represented by each category of 1-7 cores and provide further interpretation of the figure in the legend or body.

Response: We have added the related description for current legend of Fig. 2.

13. If only contigs with all 7 cores are being considered, then this raises the question of

whether longer assemblies reflect poor trimming of host sequences (with potential downstream consequences for AMG analyses). This point requires clarification.

Response: We acknowledge that there may be contamination from host organisms in some metagenome-derived viral contigs. However, in our humble opinion, this has been a challenging issue in viromics, as there is no clear boundary between bacterial and viral genome content. In our dataset, we have indeed observed some N4-like UViGs that are larger than 100-kbp and possess direct terminal repeats (DTR), indicating that these sequences were assembled as complete entities. However, it is still possible that there may be chimeric sequences or artifacts introduced during the assembly process. We have taken rigorous quality control measures to generate the sequences, but unfortunately, we are unable to fully address this issue at present. We apologize for this.

14. Fig. 4 Figure legend: What are "floated genera" (L881)? If this refers to other genera that are currently pending as taxonomy proposals please provide a reference. Panel B: What do the yellow and green colorings by the leaves indicate? The legend labeled "Branch" intermingles branch line style with individual leaf coloring. Please name the outgroup.

Response: We have provided the tree files of N4-like viral terminase (Extending data 2), the readers can retrieve all related outgroup and check their taxonomic status. The annotations related to "Branch" in the figure have been appropriately modified. We cited related literature in main txt (The "Results and Discussion" section: "Evolutionary of N4-like viruses").

15. Fig. 5 In analyses relating to the multigene phylogeny in relation to other gene phylogenies, was gene-length controlled for (to prevent longer genes from dominating the signal)? Panel B/C: The quantitative basis for defining the red boxes is not clear and I suggest omitting them; clarify that the "Top ten subfamilies" are the top ten most abundant (presumably?); however, it is not clear why all subfamilies would not have been labeled.

Response: The branch lengths in this phylogenetic tree were intentionally ignored to emphasize the clustering information rather than evolutionary distances. We have retained the boxes in the phylogenetic tree but used different colors to represent them, aiming to highlight the nearly identical clades in this co-phylogeny analysis. The figure and its

legend have been revised accordingly.

16. Though the idea of including a figure to facilitate interpretation is a nice one, it is not clear how to interpret the cartoon.

Response: We have optimized the figure and corrected the incorrect components indicated in the virion cartoon, such as the terminase large/small subunits. We have also added a more detailed description in the figure legend.

17. Fig. 7 Piecharts: The slice for "Others" is colored green as a Class I AMG, however the expanded view shows that it contains Class I and Class II AMGs, if this is the case then a third color should be used for "other"; also "Cell circle" should be "Cell cycle", here and throughout (e.g. Table 1, L265, L266). Cell cartoon: The intent to map AMGs to the cell to highlight processes is nice, however it is a bit challenging to interpret some aspects of this figure as there is a lot of information and not all the components are labeled and for some of the labels it is not known whether the cartoon represents an example of a possible function (e.g. Nutritional limitation SF64), streamlining some pieces here would be helpful.

Response: We have corrected the incorrect colors in the pie chart, and now it displays the correct information. We have also streamlined the metabolic pathway diagram associated with viruses, showing only the top five pathways involved in three categories of viruses (N4-like HQ-UViGs, N4-like viral isolates, and N4-like proviruses), as well as the top five unique pathways for each category.

18. Fig. S1-S6 - Recommend combining all of these genome diagrams into a single continuous multi-page figure that includes all 342 identified viruses all ordered onto a tree (perhaps the concatenated hallmark tree); this will only be readily viewable to the reader by zooming in quite a bit and scrolling up and down, but for anyone interested in this level of detail they will appreciate having everything on one page.

Response: According to the reviewer's suggestion, we attempted to combine the genome organization diagrams with the tree; however, the resulting visualization did not meet our expectations. We aimed to provide readers with a clear view where they could identify the elements contained in the main figure without the need for zooming in. We also consulted other experts who mentioned that some readers may prefer to print the article for reading

(especially older scholars who are accustomed to reading printed documents). Therefore, overcrowded figures would not be user-friendly for this group of readers.

However, based on the reviewer's suggestion, we have merged the original six supplementary figures into one, allowing readers interested in studying these genomes to conveniently compare them in the electronic version without the need for frequent switching between different figures. For those readers who need to understand the phylogenetic status of certain genomes in the genome organization diagrams, the manuscript provides detailed tree files and metadata for all viral contigs. They can index the corresponding branch in the phylogenetic tree by referring to the labels corresponding to the genomes of interest in the genome organization diagrams. We apologize for not fully adopting the reviewer's proposal, but we still believe that our considerations are not entirely unreasonable.

19. Fig. S7 - It is very helpful and nice to have this overview figure. Some additional points would benefit from clarification.

- Legend: The legend the numbers don't match the figure, e.g. the legend refers to 158 high quality UViGs including the 148 from IMG/VR and the 10 from GenBank, however there is no reference to 158 in the figure and the 10 from GenBank are not identified as HQ-UViGs in the figure so it is not clear for the reader.

Response: We have added relevant descriptions of 10 HQ-UViGs from GenBank at appropriate positions in the figure.

- Figure: I suggest adding headers on the top of the left and right sides & calling them panels A & B.

Response: The figure has been divided into two parts as reviewer suggested.

- Panel A: Suggest removing the line from "144 N4-like viral isolates" to "10 additional N4-like ...".

Response: The related description was removed.

- Panel A: Suggest changing: "length of N4-vRNAP" to "length of N4-vRNAP (req 2500aa min)", "N4-associated UViGs" to "N4-like-vRNAP-candidate UViGs" (at least 4 places); "CD-HIT: 99%..." to "remove near-identical sequences - CD-HiT...".

Response: Corrected.

- Panel A: The last section leading from the 601 to the 302 is not clear. There is no process box (green box) connecting the 601 to the 148 to explain how this number was reduced. It seems the bottom left section is meant to show the process reflecting searches with HMMs from all the hallmark genes to reduce the number from 601 to 148, if this is the case it would clearer to remove all of those boxes which are redundant with the top left of the figure and simply indicate this as a green box between the 601 and 148 with a statement with something like "search with 7 hallmark gene HMMS from 154 N4-like viral genomes from GenBank".

Response: We added a green box "co-occurrence of all six hallmark genes" between two yellow boxes.

- Panel A: I suggest coloring the boxes for all the unique sets with some new color -> "144 N4-likes", "10 additional ...", "148 HQ-UViGs", and "40 N4-like", as these are the final ones being added up.

Response: Corrected.

- In panel B: Suggest removing "free-living" from "Free-living N4 ..." to "N4-vRNAP hallmark-HMM"

Response: Corrected.

- In panel B: Why is this difference between the # of hits (8991) and the number of assemblages (460) so great? Are there many copies commonly found in each of the assemblages? Why would this be?

Response: The "cellular" has been removed from the original figure. We re-checked the raw data generated in this study and found that a significant portion of these hits are derived from viruses (we did not preclude the viral portion from the nr database). Additionally, some of these hits are derived from identical sequences (sequences from the "Identical Protein Group" are all included in this retrieval set). Therefore, the resulting 8991 is actually a highly redundant set that includes viruses, but our final result is non-redundant.

- In panel B: Suggest changing "co-occurrence of all six hallmark genes" to "Co-occurrence of all 7 hallmark genes"

Response: Corrected.

20. Fig. S8 This figure is not referenced in the manuscript and it is not clear what panel B is showing, for example why are there different numbers of iterations of HMM and how does this inform the reduction of the 601 HQ-UViGs to the 148?

Response: Most of the information depicted in this figure has already been described in the overview diagram. Therefore, we have removed this figure from the revised manuscript.

July 19, 2023

Prof. Yantao Liang
Ocean University of China College of Marine Life Sciences
No. 5 Yushan Road,
Qingdao, Shandong 266000
China

Re: mSystems00197-23R1 (**Identification of hidden N4-like viruses and their interactions with hosts**)

Dear Prof. Yantao Liang:

Your manuscript has been accepted, and I am forwarding it to the ASM Journals Department for publication. For your reference, ASM Journals' address is given below. Before it can be scheduled for publication, your manuscript will be checked by the mSystems production staff to make sure that all elements meet the technical requirements for publication. They will contact you if anything needs to be revised before copyediting and production can begin. Otherwise, you will be notified when your proofs are ready to be viewed.

If you would like to submit a potential Featured Image, please email a file and a short legend to msystems@asmusa.org. Please note that we can only consider images that (i) the authors created or own and (ii) have not been previously published. By submitting, you agree that the image can be used under the same terms as the published article. File requirements: square dimensions (4" x 4"), 300 dpi resolution, RGB colorspace, TIF file format.

We recognize that the video files can become quite large, and so to avoid quality loss ASM suggests sending the video file via <https://www.wetransfer.com/>. When you have a final version of the video and the still ready to share, please send it to mSystems staff at msystems@asmusa.org.

Sincerely,

Jack Gilbert
Editor, mSystems

Journals Department
E-mail: mSystems@asmusa.org